# Multifactor colorimetric analysis on pH-indicator papers: an optimized approach for direct determination of ambient aerosol pH

Guo Li[1], Hang Su[1*], Nan Ma[2], Guangjie Zheng[1], Uwe Kuhn[1], Meng Li[1], Thomas Klimach[1], Ulrich Pöschl[1], Yafang Cheng[1*]

[1] Max Planck Institute for Chemistry, Mainz, Germany

[2] Institute for Environmental and Climate Research, Jinan University, Guangzhou, China

\* *Correspondence to*: Y. Cheng (yafang.cheng@mpic.de) or H. Su (h.su@mpic.de)

## Abstract

Direct measurement of the acidity (pH) of ambient aerosol particles/droplets has long been a challenge for atmospheric scientists. A novel and facile method was introduced recently by Craig et al. (2018), where the pH of size-resolved aerosol droplets was directly measured by two types of pH-indicator papers (pH ranges: 0 – 2.5 and 2.5 – 4.5) combined with RGB-based colorimetric analyses using a model of G-B (G minus B) versus $pH^2$. Given the wide pH range of ambient aerosols, we optimize the RGB-based colorimetric analysis on pH papers with a wider detection range (pH ~ 0 to 6). Here, we propose a new model to establish the linear relationship between RGB values and pH: $pH_{predict} = a \times R_{normal} + b \times G_{normal} + c \times B_{normal}$. This model shows a wider applicability and higher accuracy than those in previous studies, and is thus recommended in future RGB-based colorimetric analyses on pH papers. Moreover, we identify one type of pH paper (Hydrion® Brilliant pH dip stiks, Lot Nr. 3110, Sigma-Aldrich) that is more applicable for ambient aerosols in terms of its wide pH detection range (0 to 6) and strong anti-interference capacity. Custom-made impactors are used to collect lab-generated aerosols on this type of pH paper. Preliminary tests show that, with a collected particle size range of ~ 0.4 – 2.2 μm, the pH paper method can be used to predict aerosol pH with an overall uncertainty ≤ 0.5 unit. Based on laboratory tests, a relatively short sampling time (~ 1 to 4 hours) is speculated for pH prediction of ambient aerosols. More importantly, our design of the impactors minimizes potential influences of changed environmental conditions during pH paper photographing processes on the predicted aerosol pH. We further show that the routinely adopted way of using pH color charts to predict aerosol pH may be biased by the mismatch between the standard colors on the color charts and the real colors of investigated samples. Thus, instead of using the producer-provided color charts, we suggest an in-situ calibration of pH papers with standard pH buffers.

## 1 Introduction

Aerosol particles have vital impacts on atmospheric chemistry, human health and global climate (Pöschl, 2005; Baltensperger et al., 2008; Pósfai and Buseck, 2010; von Schneidemesser et al., 2015; Shiraiwa et al., 2017). Understanding the basic physicochemical properties of aerosols can provide insights into various aerosol processes in the atmosphere and may further help to establish measures against air pollution. Aerosol acidity, usually quantified by aerosol pH, is one of the most important basic properties of liquid-phase aerosols. Aerosol pH has multiple effects on the other properties of aerosols, e.g., aerosol composition (Cheng et al., 2016), reactivity (Gao et al., 2004; Iinuma et al., 2004; Northcross and Jang, 2007), toxicity (Fang et al., 2017; Chowdhury et al., 2018), phase transition (Dallemagne et al., 2016; Losey et al., 2018) and their related climatic effects (Dinar et al., 2008; Hinrichs et al., 2016; Cai et al., 2018). It also plays a critical role during secondary organic aerosol (SOA) formation (e.g., Surratt et al., 2007; Gaston et al., 2014; Han et al., 2016) and in many other chemical processes in the atmosphere (Hennigan et al., 2015; Cheng et al., 2016; Wang et al., 2016a; Keene et al., 2004; Ahrens et al., 2012).

Despite its essential importance, currently there is few aerosol pH measurement data set available. One main reason is the small sizes (with an aerodynamic diameter range of 2 nm – 10 µm, see McNeill, 2017) of these atmospheric particles, rendering measurements of aerosol pH not as easy as for bulk solutions. Moreover, the non-conservative nature of $H^+$, i.e., $H^+$ concentrations do not scale in proportion to the dilution levels due to buffering effects and the partial dissociation of weak acids, further makes probe of aerosol pH a challenging topic (Hennigan et al., 2015). For direct measurements of aerosol pH, two types of methods have been employed: filter-based sample extraction (Koutrakis et al., 1988; Keene et al., 2002; Jang et al., 2008) and spectroscopic/microscopic analysis (Li and Jang, 2012; Dallemagne et al., 2016; Rindelaub et al., 2016; Craig et al., 2017; Wei et al., 2018). As the former method is offline, it suffers from both poor time resolution and intensive labor work (Hennigan et al., 2015). Moreover, it cannot account for the water in the aerosol droplets, and involves extraction with solvents that can shift the equilibria of present ions, leading to high uncertainties. The latter method is normally used for laboratory-generated particles with simple compositions that cannot fully represent ambient aerosols (Craig et al., 2018). Due to these limitations of direct measurements, thermodynamic equilibrium models such as ISORROPIA-II (Fountoukis and Nenes, 2007) and E-AIM (Clegg and Seinfeld, 2006b, a) have been widely used to estimate the acidity of ambient aerosol droplets, although comprehensive evaluations of the acidity are hampered by lack of observational data. Thus, developing new methods to directly measure ambient aerosol pH is imminently needed to constrain the output of thermodynamic models.

In a recent study, Craig et al. (2018) reported an intriguing way to directly measure aerosol pH using pH-indicator papers, which in the past are the most common and convenient tool to test the pH of bulk solutions. To measure aerosol pH, the generated size-resolved aqueous aerosol samples (($NH_4)_2SO_4$-$H_2SO_4$) were firstly collected on pH-indicator papers. Then the color of the samples on pH papers was analyzed quantitatively through a colorimetric image processing program (Matlab). In this way, the standard pH color chart of the indicator papers was used as a reference to finally derive the aerosol pH. In terms of aerosol sampling, Craig et al. (2018) collected aerosols generated in laboratory and from ambient air onto pH papers using a microanalysis particle sampler (MPS-3). The MPS-3 had three stages with aerodynamic diameter cutoff sizes ($d_{50}$) of 2.5 – 5.0, 0.4 – 2.5 and <

0.4 μm for stages 1, 2 and 3, respectively, enabling analysis of size-resolved aerosol pH. An interesting finding
from their measurements based on both pH papers and Raman spectroscopy was that, for systems with pH < 2,
the smaller particles (i.e., < 0.4 and 0.4 – 2.5 μm) displayed a markedly lower pH than the larger particles (i.e.,
2.5 – 5 μm). These results were attributed to ammonia partitioning and water loss caused by the increased surface-
area-to-volume ratios of smaller particles (Craig et al., 2018). The use of pH-indicator papers and the related
color processing technique introduced by Craig et al. (2018) tactfully circumvents the challenges and difficulties
in aerosol pH measurements. However, Craig et al. (2018) only reported two types of pH papers with relatively
high precision for pH measurements (one with pH range 0 - 2.5 and the other 2.5 – 4.5, see Craig et al., 2018),
whereas in the atmosphere the aerosol pH may vary in a wide range. Note that the authors indeed employed
another type of pH paper with a larger pH range from 0 to 6 for ambient aerosol sampling, unfortunately they
found that this paper was not compatible with their Matlab script for more quantitative analysis (Craig et al., 2018).
Additionally, due to the small area and various shape of different types of pH papers, collection of aerosols on
these materials is quite distinct from that on commonly used filters. The collected particles may induce a color
change only on a small spot (Craig et al., 2018), differing from the color variation on a much larger scale caused
by bulk solutions. Moreover, the environment under which aerosols are collected can indirectly affect the
measured aerosol pH: In an environment different from that the aerosols were originally in,
evaporation/condensation of water on pH papers might happen, which may further lead to changes in ion activities
and/or water dispersion/homogeneity on pH papers. Thus, to have accurate aerosol pH measurements, special
techniques/instruments need to be developed for effective aerosol collection and pH paper color recognition, and
meanwhile careful design should be made to avoid potential impacts of varied environmental factors on the
predicted aerosol pH.

The colorimetric method used by Craig et al. (2018) was based on analyzing the red (R), green (G) and blue (B)
channels of the sample images, where a linear dependence of the difference between G and B (G-B) on $pH^2$ was
found. According to trichromatic theory, RGB are the three primary colors and their combination in varying
proportions can generate any other specific color (Su et al., 2008). The standard RGB scale is represented by the
values of R, G and B, and each has a range from 0 to 255. For example, the number [0, 0, 0], i.e., R = 0, G = 0,
B = 0, corresponds to absolute black and [255, 255, 255] to true white. RGB-based image analysis has been
applied in the fields of inorganic and analytical chemistry. For instance, Selva Kumar et al. (2018) found a good
linearity between concentrations of Thorium ions ($Th^{4+}$) and the ratio of R and G (R/G), and Wan et al. (2017)
reported a relation between bovine serum albumin (BSA) concentrations and the normalized values of R, G and
B, respectively. In these previous studies, different RGB models (i.e., ways to interpret the RGB values) were
adopted, however with few detailed explanations on the intrinsic reasons. To further enhance the reliability and
comparability of the data associated with RGB analysis, a unified model/method to deal with the RGB information
is needed, especially for the pH determination of aerosols where high uncertainty of measured pH values can have
a huge impact on the pH-dependent multiphase chemical processes.

Considering that the pH values of ambient aerosols can cover a wide range (up to ~ 6) (von Glasow and Sander,
2001; Pszenny et al., 2004; Song et al., 2018; Shi et al., 2019), the goal of the present study is to optimize the
RGB-based colorimetric analysis on pH-indicator papers for direct determination of ambient aerosol pH in a wider
detection range (pH ~ 0 to 6). We thus propose a new way to analyze the RGB values and establish the relationship
between RGB and pH. We further compare our proposed RGB model with the models used in previous studies
in terms of evaluating the established linear relationship between RGB and pH. In addition, the routine way of
using a pH color chart to derive the pH of samples is inspected, and the results reveal some deficiencies of this
method. Therefore, we suggest an optimized way to use pH papers for aerosol pH prediction with higher precision
and accuracy. Nine types of pH papers are tested for their potential of probing pH of ambient aerosols. Among
these pH papers, only one type is found to be the most suitable and is further tested for its capability of predicting
the pH of lab-generated aerosols by using two custom-made impactors.

## 2 Materials and Methods

### 2.1 pH-indicator papers

Nine types of pH-indicator papers were adopted in this study. Each type has a pH color chart that is accompanied
with the pH papers and supposed to serve as a reference to quantify the pH of a sample through colorimetric
analysis. Details about the pH paper detection ranges and the corresponded type classification used in this work
can be found in Table S1. The first two types are the same as used by Craig et al. (2018), aiming to compare our
results with those from Craig et al. and validate our colorimetric image processing method. The others have larger
pH detection ranges covering the generally observed pH range of ambient aerosols. Note that in this study, we
mainly focused on the first five types of pH papers and the remaining four types were also evaluated and compared
with the first five types in terms of their resistance to chemical interference and potential capability to measure
the pH of ambient aerosols.

### 2.2 pH buffers, aerosol sample solutions and lab-generated aerosols

To examine the correlation between RGB and pH, eight standard pH buffer solutions were used as purchased and
meanwhile several other buffers (with different pH values as the purchased ones) were obtained by mixing the
commercial buffers with solutions of sodium hydroxide or hydrochloric acid (prepared using de-ionized water,
18.2 MΩ cm). pH values of all the buffers were further checked by a pH bench meter (model: HI 2020-02, Hanna
Instruments Inc., USA). Prior to the check, the pH meter was calibrated with a three-point calibration mode using
the standard buffer solutions provided by Hanna Instruments Inc., USA. The measured pH values and their
standard derivations are listed in Table S2, and the measured pH values show a small deviation from those
specified on the buffer solution bottles, within the displayed uncertainties concomitant with these specified values.
Considering that some inorganic/organic components of ambient aerosols might interfere with the dyes on pH
papers and cause biased estimation of pH, salt systems with varying inorganic and/or organic acids common in
aerosols and pH levels (as measured by the pH bench meter) were employed to test the applicability of different
types of pH papers combined with our RGB model. Details about the composition of the tested salt systems can
be found in Table S3. In general, the inorganic systems were similar to those used by Craig et al. (2018). Here,
we further tested the influence of organic acids on pH paper performance by adding organic acids into the
inorganic systems. As oxalic acid ($C_2H_2O_4$) and malonic acid ($C_3H_4O_4$) were frequently detected in tropospheric
aerosols and found to be the dominant short dicarboxylic acids in aerosol composition (Abbatt et al., 2005;
Falkovich et al., 2005), they were adopted in this study. For the solution preparation of each system, varying
amounts of 1 M inorganic/organic acids were added into 30 mM inorganic salt solution to achieve different pH
levels (Surratt et al., 2008; Craig et al., 2018).  To prepare the inorganic and organic mixtures, the amount of
added organic acids was generally two times larger than the inorganic acids and the final salt concentration could
be as low as 15 mM due to the dilution effect of added acids.  To prepare the solutions, all chemicals were used
as purchased: NaOH ($\geq$ 99.0%, Roth, Germany), $Na_2SO_4$ ($\geq$ 99.0%, Merck, Germany), $NaNO_3$ ($\geq$ 99.0%, Merck,
Germany), $Na_2CO_3$ ($\geq$ 99.5%, Sigma-Aldrich, USA), $(NH_4)_2SO_4$ ($\geq$ 99%, Sigma-Aldrich, USA), $NH_4NO_3$ ($\geq$
98.0%, Fisher Chemical, USA), $MgSO_4$ (> 98%, neoFroxx GmbH, Germany), $H_2SO_4$ (98%, Merck, Germany),
$HNO_3$ (65%, Merck, Germany), HCl (37%, Merck, Germany), $C_2H_2O_4 \cdot 2H_2O$ ($\geq$ 99%, Sigma-Aldrich, USA) and
$C_3H_4O_4$ (99%, Sigma-Aldrich, USA).

To test the feasibility of the colorimetric analysis method towards real aerosols, the prepared aerosol sample
solutions (i.e., the inorganic and organic mixtures) were further used to generate aerosol particles through an
aerosol generator under laboratory conditions.  The lab-generated aerosols were collected onto the type V pH
paper through two custom-made impactors, which had different cutoff sizes and were connected in series.  Before
collection, the nebulized aerosols were firstly mixed with humidified and HEPA-filtered air to reach a relative
humidity (RH) of 90 $\pm$ 1.5% and a total flow rate of 28.6 L $min^{-1}$.  To minimize water exchange between the
generated aerosol flow and the humidified aerosol-free air flow, the RH of the air flow was maintained similar to
that of the aerosol flow.  With the sampling flow rate of 28.6 L $min^{-1}$, the upstream impactor had a cutoff diameter
($d_{50}$) of ~ 2.2 $\mu$m (identified by an UV-APS, model 3314, TSI Inc.) and the downstream impactor had a $d_{50}$ of ~
0.40 $\mu$m (identified by a SMPS, model 3082, TSI Inc.).  These two impactors produced a total pressure drop of
57 mbar in the aerosol line (measured by a digital pressure meter, model GMH 3111, GHM Messtechnik GmbH,
Germany).  To validate our method, one wifi endoscope camera was installed on the top of the downstream
impactor (with a collected particle size range of 0.40 – 2.2 $\mu$m) to capture the images of one pH paper (5 $\times$ 5 mm)
fixed on the impactor bottom plate.  In practice, we could install a camera for each impactor.  In order to apply our
RGB model (Sect. 2.4), a series of standard buffers were also adopted to generate aerosols with the same
experimental configuration mentioned above.

Given that in real ambient case some light-absorbing particles, such as black carbon (BC), may interfere with the
displayed color of pH papers and thereof cause biased pH prediction, commercial soot samples (fullerene soot,
Lot Nr. L20W054, Alfa Aesar, Germany) were additionally mixed into the aerosol sample solutions for aerosol
generation to check their potential impact on the predicted aerosol pH.  To achieve that, pure BC suspension was
firstly prepared with de-ionized water and then a 15-minute ultrasonic treatment was performed to enhance the
dispersion of BC particles inside the suspension.  The mass concentration of BC particles (measured under dry
conditions with a RH = 14%) generated from this suspension was quantified by the SMPS as ~ 240 $\mu$g $m^{-3}$ using
the density of fullerene soot of 1.72 g $cm^{-3}$ (Kondo et al., 2011).  5 mL of this suspension was additionally mixed
into 10 mL of pre-prepared aerosol sample solution, and this mixture was finally used for aerosol generation.  A
total mass concentration of the generated aerosols (measured under dry conditions with a RH = 14%) was
determined by the SMPS as ~ 800 $\mu$g $m^{-3}$ using a density of 1.7 g $cm^{-3}$. This density was obtained by averaging
the densities of different components weighted by their respective volume in the aerosol sample solution mixed
with BC. Note that the BC mass fraction was ~ 10%, representing a typical BC contribution in ambient aerosols
(Wang et al., 2016b; Chen et al., 2020).
**2.3 Correlation between RGB and pH**
Figure 1 shows the procedure of how to use a colorimetric analysis to obtain the correlation between RGB and
pH. First, 2 μL of liquid samples was dripped onto each piece of pH paper held by a clean transparent glass plate
(with the other side coated by a piece of graph paper). This adopted small volume (2 μL) was based on a general
estimation of the available amounts of liquid aerosols for aerosol sampling under a typically polluted conditions
(with $PM_{2.5}$ mass concentration around 100 μg m$^{-3}$) with high RH (60% – 80%), and assuming an aerosol
collection efficiency of 50% and a sampling flow rate of several hundred liter per minute (e.g., can be achieved
by a Tisch Environmental $PM_{2.5}$ high volume air sampler, see https://tisch-env.com/high-volume-air-
sampler/pm2.5) with a sampling time of a few (2 - 4) hours. Here, the used $PM_{2.5}$ mass concentration and RH
refer to the conditions during haze events which are frequently occurring in China. For example, during the most
severe haze episodes in January 2013, monthly averaged $PM_{2.5}$ concentration in Beijing reached 121 μg m$^{-3}$ and
the RH was constantly at a level of 60% – 80% (Zheng et al., 2015). Even the air quality in China has significantly
improved in recent years, the number of days with moderate haze (with daily mean $PM_{2.5}$ concentration in the
range of 100 – 200 μg m$^{-3}$) in the North China Plain shows on obviously decreasing trend from 2004 to 2018 with
an average of 113 d (Zhang et al., 2020). Note that, we further estimated the minimum sample volume and mass
needed to generate a measurable color change on the suggested pH paper. The related results are shown below.
Then an image of the sample was captured by a smartphone camera (Apple iPhone 5s in this study) immediately.
Similar to Craig et al. (2018), the corresponding color chart of the used pH paper was included into each image
to correct for potential influences of variations of light source and angle during photographing. The digital images
were processed by an Adobe Photoshop software to crop a square with $100 \times 100$ pixels at the center of the sample
(as well as each color chip on the color chart). The RGB information of the cropped square was then obtained
and further analyzed by Matlab (The MathWorks, Inc. version R2018b).
**2.4 RGB model**
Considering that a color is represented by combination of R, G and B values, a linear combination of these three
primary colors should be able to reflect the characteristics of the color and therefore represent the pH related to
the color. Su et al. (2008) reported a good correlation between the linearly combined RGB and the contents of
chlorophyll *a* and lipid, respectively in microalgae. To further account for the effect of changing light intensity
on the obtained RGB values, each color channel should be normalized at first (Yadav et al., 2010). The
normalization can be achieved through Eqns. (1) – (3) shown below:

$R_{\text{normal}} = R/(R + G + B)$         (1)
$G_{\text{normal}} = G/(R + G + B)$         (2)
$B_{\text{normal}} = B/(R + G + B)$         (3)

where $R$, $G$ and $B$ are the mean value of each primary color on the entire $100 \times 100$ pixels image, respectively. Note that every pixel has an RGB value vector: [R, G, B]. Then a model describing the linear combination of RGB can be given as follows:

$$pH_{\text{predict}} = aR_{normal} + bG_{normal} + cB_{normal} \qquad (4)$$

where the linear combination $aR_{normal} + bG_{normal} + cB_{normal}$ essentially represents the color information and here can be treated as equivalent to the predicted pH ($pH_{\text{predict}}$) based on RGB analysis; $a$, $b$ and $c$ are the coefficients, which can be determined by linear regression analysis through Matlab. The linear regression function is expressed as:

$$Y = aX_1 + bX_2 + cX_3 \qquad (5)$$

where $Y$ is the dependent variable vector, $X_1$, $X_2$ and $X_3$ are independent variable vectors. These vectors can be achieved from a standard color chart or a series of buffer samples (with known pH values) on pH papers: $Y$ is the series of pH values (i.e., reference pH, $pH_{\text{reference}}$) shown on the color chart or of buffer samples (as shown in Fig. 1, the pH papers with different pH buffer solutions are collected together to form a pH series); $X_1$, $X_2$ and $X_3$ are the normalized average of R, G and B respectively, based on analysis on the detected colors. As a color chart is normally used as a reference for pH measurements using pH papers, a linear regression analysis on the color chart can provide the coefficient vector [$a$, $b$, $c$] as an answer. Then the same set of coefficient vector (i.e., [a, b, c]) are used to predict the pH (i.e., $pH_{\text{predict}}$) of samples using Eqn (4). Thus, with this RGB model, a linear relationship between RGB-predicted pH ($pH_{\text{predict}}$) and reference pH ($pH_{\text{reference}}$) is expected for the calibration (as shown in Fig. 1), in order to finally predict the sample pH on a pH paper.

## 3 Results

### 3.1 Validation of the new RGB model

As the RGB model (i.e., G-B vs pH$^2$) used by Craig et al. (2018) produced good linear correlations for the two types of pH papers that the authors adopted, we first examined the validity of this RGB model against the first five types of pH papers used in this work. Note that here the first two types of pH papers are the ones used and recommended by Craig et al. (2018). Figure S1 shows the relationship between average G-B and pH$^2$ derived from the color charts of these five types of pH papers, respectively. Relatively good linear correlations can be found for the first three types, which is consistent with Craig et al. (2018). However, non-monotonic correlations are encountered for the last two types of pH papers, which are the ones with wider pH detection ranges ($0.5 - 5.5$ and $0 - 6$, respectively). These results indicate a limited feasibility of the RGB model proposed by Craig et al. (2018).

Thus, the validity of our new RGB model was further checked through the five types of pH papers. The colors on the color chart for each type of pH paper were firstly analyzed through our RGB model and then the calculated $pH_{\text{predict}}$ were compared with the reference pH shown on the color chart. As shown in Fig. 2a-e (the 'color chart'

column on the left-hand side), good linearity between $pH_{predict}$ and $pH_{reference}$ can be observed for all these pH paper
types.
As aforementioned, besides the RGB model used by Craig et al. (2018), other models have also been adopted to
create a linear correlation between RGB and concentrations of the chemicals of interest in previous colorimetric
analyses (Su et al., 2008; Yadav et al., 2010; Wan et al., 2017; Selva Kumar et al., 2018). However, few
comparisons have been made regarding the goodness of the established linearity using these RGB models. Here
we further compared our model with the other two models (i.e., R/G vs pH and G-B vs $pH^2$) proposed by Selva
Kumar et al. (2018) and Craig et al. (2018) respectively, in terms of evaluating their correlation coefficient, $R^2$.
Figure S2 (the 'color chart' panel on the left-hand side) displays the $R^2$ of the established linear correlation between
$pH_{predict}$ and $pH_{reference}$ when the three models are used for the five types of pH papers, respectively. For the color-
chart-derived linear correlation, the model G-B vs $pH^2$ presents poor goodness-of-fit for type IV and V pH papers
(consistent with the results shown in Fig. S1). The model R/G vs pH shows relatively high $R^2$ for all the pH paper
types. Nevertheless, this RGB model still underperforms our model. Overall, our RGB model could provide a
high $R^2$ (> 0.95) for all the five types of pH papers, which demonstrates the universal validity of our RGB model.
**3.2 Calibration with standard buffer solutions**
A good linearity, however, may not always be obtained from the color chart of some types of pH papers in some
pH ranges. For example, in the 'color chart' column of Fig. 2, the $pH_{predict}$ present small but discernable deviations
from $pH_{reference}$ for types I, III and IV pH papers. And the type V pH paper shows even larger differences at both
ends of the pH range. Similar phenomenon was also observed in the study of Craig et al. (2018) with their RGB
model, where they argued that the pH paper dye became less effective at the limits of the pH paper range, due to
the $pK_a$ values of the dye were normally at the middle of the pH range. But it may also originate from some color
bias due to the differences between the experiment conditions and the ones under which the color chart is made
by the producer.
Thus, following the same procedure as for the color chart (see Fig. 1), pH papers with samples of a series of 2 μL
standard buffer droplets were also measured. The pH values of the standard buffers were known beforehand and
further checked with a pH meter (also denoted as '$pH_{reference}$', see Table S2). Figures 2f-j (the '2 μL buffer' column
on the right-hand side) show the comparison between $pH_{predict}$ and $pH_{reference}$ for the samples of 2 μL buffers. Much
better linearity between $pH_{predict}$ and $pH_{reference}$ can be observed for all the five types of pH papers. Especially, the
significant deviation of $pH_{predict}$ from $pH_{reference}$ found in the left panel (the 'color chart' column) disappear for the
type I and V pH papers. This means that the deviations at the edge of the pH range in the color-chart-derived
calibration curves are mainly due to the color bias of the color chart itself or caused during photographing.
Actually, even small deviations found in the color-chart-derived calibration curves (the 'color chart' column in
Fig. 2) may lead to significant or non-negligible errors in measuring aerosol pH. We conducted a case study using
the type IV pH paper combined with our RGB model to predict the pH of buffer samples by using the color-chart-
derived coefficient vector [$a$, $b$, $c$], i.e., the color-chart-calibration method (Fig. 2d). The blue symbols in Fig. S3
represent $pH_{predict}$ versus $pH_{reference}$ of the standard buffer samples. Systematical underestimation of $pH_{predict}$ can
be found at the lower $pH_{reference}$ values (i.e., $pH_{reference} = 1$, 1.68 and 2) but a slight overestimation is observed at
$pH_{reference} = 5$. This significant discrepancy may be attributed to the mismatch between the reference colors on the
color chart and the real colors of the samples, due to the differences between our experiment conditions and the
ones under which the color chart is made by the producer. This gives us a hint that the coefficient vector derived
from the color chart is not suitable for predicting the pH of aerosol samples.

For the established linear relationship using 2 µL standard buffers, the performances of different RGB models
were further compared and the results are shown in Fig. S2 (the '2 µL buffer' panel on the right-hand side). Our
RGB model still outperforms the other two models for all the five types of pH papers employed in this work.
Overall, the good agreement between $pH_{predict}$ and $pH_{reference}$ for all these tested pH papers verifies the wide
applicability of our RGB model to the pH paper calibration using standard buffers. In the following section, we
will examine the quality of predicting samples pH with the standard-buffer-calibration method.
**3.3 pH estimation for aerosol surrogates ((NH$_4$)$_2$SO$_4$-H$_2$SO$_4$) with the type IV and V pH papers**
In order to test the feasibility of pH papers with larger pH detection ranges for pH prediction of aerosols, we
further used the type IV and V pH papers to estimate the pH of lab-prepared aerosol surrogates ((NH$_4$)$_2$SO$_4$ -
H$_2$SO$_4$). To minimize the effect of varying photographing conditions (e.g., angle, light variation) on the colors of
the captured image, experiments were carried out in a cupboard with a constant light source. In addition, the pH
paper samples as well as the smartphone were fixed on a small glass plate and a metal holder respectively, to keep
their position unchanged throughout the experiment. Note that applying the standard-buffer-derived coefficients
(Fig. 2i and 2j) for pH prediction of samples required the same constant light source conditions for sample imaging
processes as for standard buffers.

$pH_{predict}$ versus $pH_{reference}$ for the 2-µL-droplet samples on the type IV pH paper are shown in Fig. S4. Generally,
the $pH_{predict}$ by the type IV pH paper are comparable with the $pH_{reference}$ at a lower pH range (i.e. $pH_{reference} = 0.46$,
1.52 and 3.0). However, an anomalous point (highlighted by the arrow in Fig. S4) with 1.5 unit of overestimation
in $pH_{predict}$ can be found at $pH_{reference}$ around 4. This overestimation was proved to be reproducible by our six
replicate experiments and it was even found for samples of diluted H$_2$SO$_4$ solutions with reference pH around 4
on the type IV pH paper. Such overestimation may be due to the chemical interferences caused by the samples
or the low buffering levels of the samples. Thus, the type IV pH paper is not recommended for future pH
measurements of aerosols. However, it may still work well for the other sample types, such as found for our self-
prepared phosphate buffers (Fig. S4). On the other hand, the type V pH paper shows decent agreements between
$pH_{predict}$ and $pH_{reference}$ within the examined pH range, as shown in Fig. 3a. Moreover, the $pH_{predict}$ are also
compared with the results by Craig et al. (2018). The orange and blue bars in Fig. 3a represent the measured pH
ranges for the lab-generated (NH$_4$)$_2$SO$_4$ - H$_2$SO$_4$ aerosols with particle sizes larger than 2.5 µm using pH papers
(the same as the type I and II pH papers used here) and Raman spectroscopy, respectively.
**4 Discussion**

**4.1 Chemical interference**

As aforementioned, aerosol samples with different compositions may have interferences on the indicating color of a pH paper and thereby cause its poor performance for aerosol pH prediction, e.g., the overestimation of pH of aerosol surrogates $((NH_4)_2SO_4\text{-}H_2SO_4)$ with the Type IV pH paper. To test the capability of chemical resistance of the Type V pH paper, we further tested its performance of predicting the pH of lab-prepared aerosol surrogates with varying inorganic/organic compositions commonly exist in ambient aerosols.

Figure 4 displays $pH_{predict}$ versus $pH_{reference}$ for our lab-prepared droplet samples (2 µL) under different pH levels using the Type V pH paper. As shown in Fig. 4a, systematic divergences between $pH_{predict}$ and $pH_{reference}$ (i.e., overestimation of $pH_{predict}$ when $pH_{reference}$ is in the range of 2.5 – 3.5 whereas underestimation of $pH_{predict}$ when $pH_{reference} > \sim 4.5$) can be found for these tested inorganic systems. Interestingly these mismatches disappear when the organic acids are introduced into these inorganic systems (Fig. 4b), and also for the cases when the inorganic acids are replaced by organic acids (Fig. S5). In Fig.4b, this good agreement for $pH_{predict}$ versus $pH_{reference}$ is observed not only for systems containing oxalic acid ($C_2H_2O_4$, solid markers) but also for those having malonic acid ($C_3H_4O_4$, hollow markers) with an average deviation (of $pH_{predict}$ from $pH_{reference}$) < 0.5 unit. The fact that the existence of organic acids significantly improves the quality of $pH_{predict}$ may be attributed to the enhanced buffering levels of the samples (Fillion et al., 1999; Li et al., 2016). Actually, good agreement between $pH_{predict}$ and $pH_{reference}$ is found for both the inorganic and organic phosphate systems (Fig. S6) based on our further tests, which is probably due to the high buffering levels of these systems maintained by the phosphate itself (Hourant, 2004). Nevertheless, the solvent effect of the added organics on acid dissociation equilibria may also play a role (Padró et al., 2012). The detailed mechanisms may need to be explored in future studies. Given the large contribution of organics (Jimenez et al., 2009) and the well-known dominance of both organic acids (i.e., oxalic acid and malonic acid) in ambient aerosols (Abbatt et al., 2005; Falkovich et al., 2005), the potential interferences found for the inorganic systems can be expected to become vanished when organics are concomitant under ambient conditions. Additionally, the interference check was also performed for the other pH paper types (type III and VI-IX) that have larger pH detection ranges. Similar to the type IV pH paper, significant deviations of $pH_{predict}$ from $pH_{reference}$ ($\geq$ 1.5 unit) were observed for these types (see SI text and Fig. S7).

**4.2 Black carbon (BC) interference**

To apply the pH paper method to ambient aerosols, another potential interference on the captured pH paper color would come from some light-absorbing aerosols such as black carbon (BC) or brown carbon (BrC). Therefore, we further examined the potential interference of BC on the predicted pH of lab-generated aerosols. Details regarding the aerosol generation and collection can be found in Sect. 2.2.

Figure 5 shows $pH_{predict}$ versus $pH_{reference}$ for the generated aerosol particles (i.e., $(NH_4)_2SO_4\text{-}H_2SO_4\text{-}C_3H_4O_4$) with and without the co-existence of BC. Note that $pH_{reference}$ refers to the pH of bulk solutions used for aerosol generation. Generally, within the examined pH range no significant difference can be found between the $pH_{predict}$ of aerosols with BC and that of the aerosols without BC. The linear fitting (i.e., the orange and blue dashed lines in Fig. 5) for each type of dataset shows that the $pH_{predict}$ for aerosols with BC is slightly lower than the samples without BC at the low pH side but an opposite trend can be found in the high pH side. This statistically small

difference is further confirmed by running two-sample $t$-tests with Matlab, as shown in Table S4. Even this
difference ($\leq 0.5$ unit) is slight and acceptable, it indicates the existence of potential interferences of BC on the
predicted aerosol pH, and related mechanisms may need to be explored in future studies. Note that for our lab
experiments the adopted BC amount accounted for ~ 10 % of the total aerosol mass, which reflects the typical BC
contributions in ambient aerosols (Wang et al., 2016b; Chen et al., 2020).

Moreover, both types of aerosols display a lower $pH_{predict}$ than $pH_{reference}$ in the low pH range as $pH_{reference} < 2.5$
(Fig. 5). Within the same lower pH range, significantly reduced aerosol pH (versus the pH of bulk solutions)
predicted by both pH papers and Raman spectroscopy were also found in Craig et al. (2018) for lab-generated
aerosols, as indicated by the neighbored orange and blue bars in Fig. 5. Their results (Craig et al., 2018) further
revealed that the markedly-lower-$pH_{predict}$ trend weakened at the higher pH range (i.e., $2.5 < pH_{reference} < 4.5$, see
the orange bars in Fig. 5). The authors argued that the decreased aerosol pH found for smaller-size particles (with
aerodynamic diameter < 2.5 μm) could be attributed to ammonia partitioning and water loss (Craig et al., 2018).
Even with controlled RH for the aerosol dilution air flow in this study (Sect. 2.2), we cannot totally exclude the
impact of water loss on the predicted aerosol pH, considering that under such a high RH (~ 90 %) a small
difference between the RH of the generated aerosol flow and that of the dilution flow may cause non-negligible
water exchange between aerosols and the carrying gas.

In addition, the results shown in Fig. 5 further demonstrate the technical feasibility of using our custom-made
impactors for aerosol collection. More importantly, with this impactor setup, we could monitor the change of the
pH paper color at any sampling time without interrupting the sampling. Thus, when used for future ambient aerosol
collection we would expect a small difference between the surrounding environment of aerosols inside the
impactors and ambient conditions.

### 4.3 Identification of the needed minimum sample amount and sampling time for the type V pH paper

The pH of ambient aerosols can be changing due to the varying atmospheric composition (e.g., some important
trace gases like $SO_2$, $NO_2$, $NH_3$ and organic acids) and physical characteristics (e.g., ambient relative humidity
(RH) and temperature (T)). Thus, reflecting the temporal evolution of aerosol pH requires a pH measurement
method with a high time resolution. As aforementioned, to collect 2 μL of liquid aerosol samples, a sampling
time of 2 – 4 hours is needed even using a high-volume air sampler with a sampling flow rate of several hundred
liter per minute. Here, in order to have a higher time resolution and/or a lower sampling flow rate, we further
identified the minimum sample volume needed to generate a measurable color change on the type V pH paper.
Figure 3b shows the results for 0.1μL of lab-prepared aerosol sample solutions. Similar to the RGB analysis
procedure used for the 2 μL samples (e.g., in Fig. 3a), the $pH_{predict}$ in Fig. 3b are calculated with the averaged
coefficient vector [$a$, $b$, $c$] derived from three replicate calibration experiments with 0.1 μL standard buffers (Fig.
S8). Generally, $pH_{predict}$ agrees well with $pH_{reference}$, with biases (averaged $pH_{predict}$ versus $pH_{reference}$) within 0.5
unit. Note that these experiments were carried out under laboratory conditions with a relatively stable RH of 50
$\pm 1$ % and T of $23 \pm 1$ ºC. To avoid fast water exchange between the lab air and our samples as well as potential
interfering effects (absorption/reaction) caused by the lab air, the 0.1 μL samples were transferred (through a
pipette) directly onto the pH paper surface and each sample was immediately photographed ($\leq$ ~ 3 seconds) after
it got contact with the pH paper dye.  Due to this extremely small sample volume, the influence of lab air on
sample pH could become prominent because a significant sample color change was frequently observed after the
sample was exposed to the lab air for > ~ 5 seconds.

This tiny volume corresponds to a sample mass of ~ 180 μg assuming an effective density of 1.8 g cm$^{-3}$ for ambient
aerosols (Sarangi et al., 2016; Geller et al., 2006), which is comparably low to the needed minimum particulate
masses in Craig et al. (2018), i.e., ~ 65 μg to ~ 2.5 mg for PM$_{2.5}$ or larger particles with pH from 0 - 2.5 to 2.5 - 4.
Note that, as we pipetted 0.1 μL (~ 180 μg) samples on the type V pH paper, this amount cannot be directly used
for estimations of the time needed for ambient aerosol sampling, which also depends on how aerosols will be
collected on the pH paper.  However, this minimum-sample-amount test could provide us a general estimation on
the lower limit of the needed volume/mass of collected ambient aerosols, which can further guide us to search for
new techniques/instruments for aerosol collection as well as color recognition.   As described in Sect. 2.2, two
custom-made impactors were employed to collect lab-generated aerosols on the type V pH paper.  Since sampling
time determined the amount of collected aerosols and thereby affected the displayed color on pH papers, an
optimal sampling time of 30 minutes was identified in this study by examining the established linearity between
$pH_{predict}$ and $pH_{reference}$ (with the Eqn (5)) for aerosols generated from standard buffers.  With this sampling time, a
good linearity with R$^2$ > 0.95 was established (Fig. S9).  Taking the sampling time and the mass concentration of
lab-generated aerosols (~ 800 μg m$^{-3}$, measured under RH = 14%) into account, we would infer a sampling time
of ~ 4 hours will be needed to generate one predicted aerosol pH for ambient aerosols under typically polluted
conditions (with PM$_{2.5}$ mass concentration around 100 μg m$^{-3}$).  Since this time estimation is based on a sampling
flow rate of 28.6 L min$^{-1}$, future samplings with a higher time resolution (e.g., ~ 1 hour) probably can be achieved
by adopting a much larger sampling flow rate (e.g., ~ 120 L min$^{-1}$).  Our preliminary tests have indicated that the
cutoff size, collection efficiency and pressure drop of the impactors strongly depended on the sampling flow rate
and flow direction (i.e., switch between inlets and outlets of the impactors).  More characterizations on the
impactors will be done in our future work.

These results confirm the feasibility of the type V pH paper as well as our RGB model for pH estimation of the
aerosol sample solutions with a volume even down to 0.1 μL and of the lab-generated aerosols collected by
impactors.  The speculated low sampling time (i.e., ~ 1 hour) at high sampling flow rates and the large pH detection
range of the type V pH paper highlight its potential for future development of real-time aerosol pH measurements.
Moreover, instead of using a color chart to calibrate pH papers for each sample (Craig et al., 2018), our results
demonstrate that the in-situ calibration method of using standard buffers as well as standard-buffer-generated
aerosols (independent of different samples) can derive an averaged coefficient vector [*a*, *b*, *c*] which can be
uniformly applied to pH prediction of different samples provided the photographing conditions are kept constant.
This unique feature further facilitates the application of the type V pH paper under ambient cases.
**5 Conclusions**
We proposed a new model to establish the correlation between the color of droplet/aerosol samples on pH-
indicator papers and their measured pH.  The model was based on RGB analysis of the images of samples.
Comparison of our model and another two RGB models verified the high reliability of our model.  Using our RGB

model, good agreement between the model-predicted pH ($pH_{predict}$) and reference pH ($pH_{reference}$) for pH paper color charts as well as standard buffers were observed for all the tested types of pH papers. Different types of pH papers with larger pH detection ranges were further examined for their performance to predict the pH of 2-µL-droplet samples with varying inorganic/organic compositions common in ambient aerosols. Only the type V pH paper (with a pH range of ~ 0 - 6) performed well and therefore was further used to estimate the pH of lab-generated aerosols. The results showed that, even under the potential interference of BC, the type V pH paper could derive aerosol pH with an uncertainty within 0.5 unit, suggesting that it deserves practical applications for pH measurements of ambient aerosols. The minimum liquid sample mass/volume needed for the type V pH paper was identified as ~ 180 µg/0.1 µL. And the current-stage tests on aerosol collection and pH estimation under lab conditions helped to infer an ambient sampling time of ~ 4 hours will be needed for typically polluted conditions, which, however, probably could be further improved to ~ 1 hour by using a high sampling flow rate. Whereas, the other pH paper types might suffer from some chemical interferences during pH measurements and therefore could generate large biases for the measured pH of aerosols. The routine procedure of using pH papers to estimate a sample pH was also examined in a case study using the type IV pH paper. The results showed that referring to the color chart for pH estimation (i.e. the color-chart-calibration method) might cause a bias of the predicted pH. To use the pH papers in a more proper and accurate way, here we further demonstrated that the in-situ calibration method of using standard buffers and standard-buffer-generated aerosols (independent of different samples) could derive an averaged coefficient vector [$a$, $b$, $c$], which can be uniformly applied to pH prediction of different droplet and aerosol samples provided the photographing conditions are kept constant.

**Data availability**

The underlying research data and Matlab code can be accessed upon contact with Guo Li (guo.li@mpic.de), Yafang Cheng (yafang.cheng@mpic.de) or Hang Su (h.su@mpic.de).

**Supplement**

The supplement is available in a separate file.

**Author contributions**

Y.C. and H.S. conceived and led the study. G.L. performed experiments and data analysis. Y.C., H.S., U.P., U.K., N.M., G.Z., M.L., T.K. discussed the results. G.L. and Y.C. wrote the manuscript with inputs from all co-authors.

**Competing interests**

The authors declare that they have no conflict of interest.

**Acknowledgement**

We acknowledge the National Natural Science Foundation of China (grant no. 91644218) and the National Key Research and Development Program of China (grant no. 2017YFC0210104). This study was supported by the Max Planck Society (MPG). G. L. acknowledges the financial support from the China Scholarship Council (CSC). G. L. also would like to thank Ping Zhang, Jinqian Zhai and Yunkun Lang for their very helpful discussions concerning the GRB model development.

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

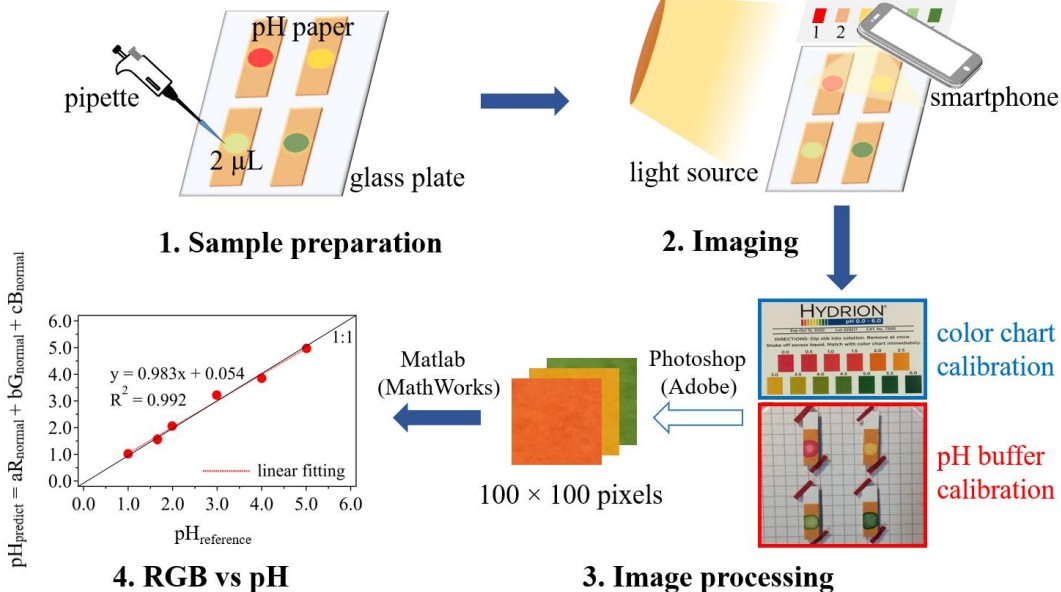

**Figure 1:** Schematic of using the RGB-based colorimetric method for pH estimation. For the color-chart-calibration method,
both the color chart and the standard buffer samples are imaged into one digital photo for subsequent processing. For the
standard-buffer-calibration method, only the standard buffer samples are used for imaging. Note that when using the standard-
buffer-calibration results to predict the pH of aerosol samples, the photographing conditions for the samples are the same as
those of the buffer calibration.

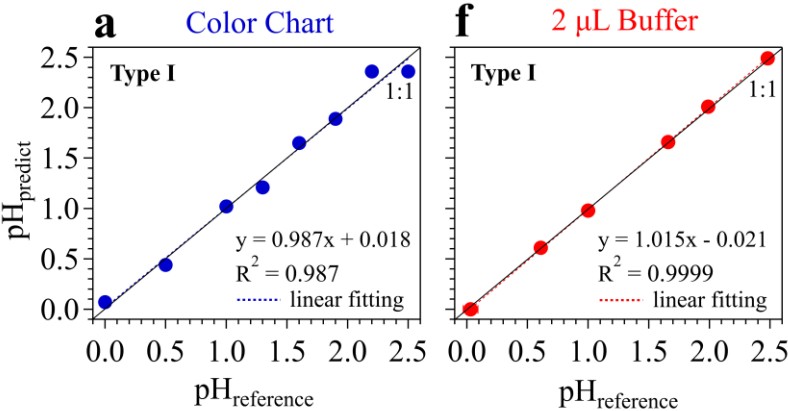


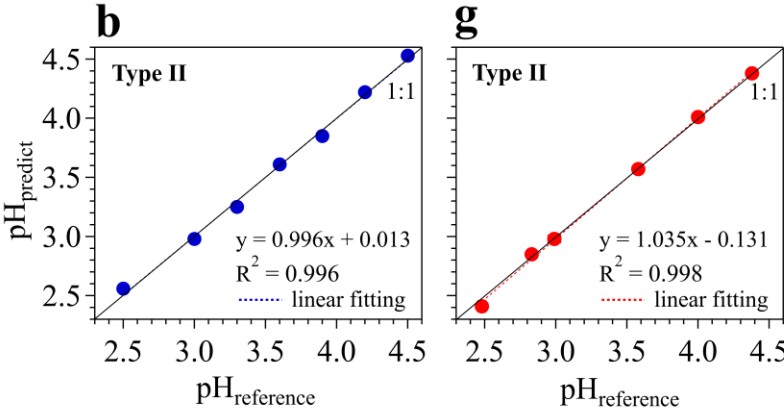


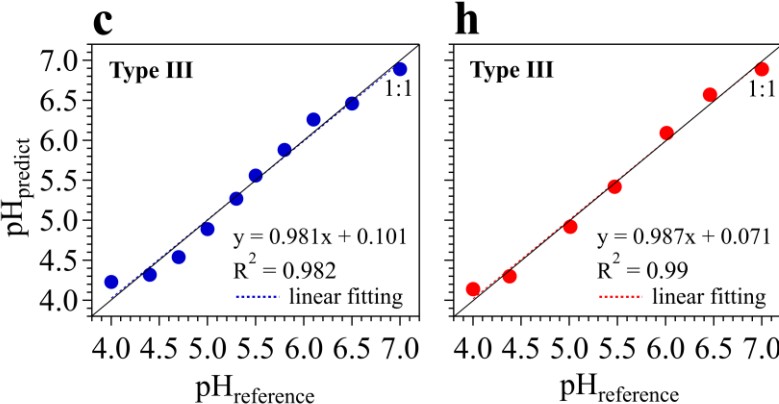


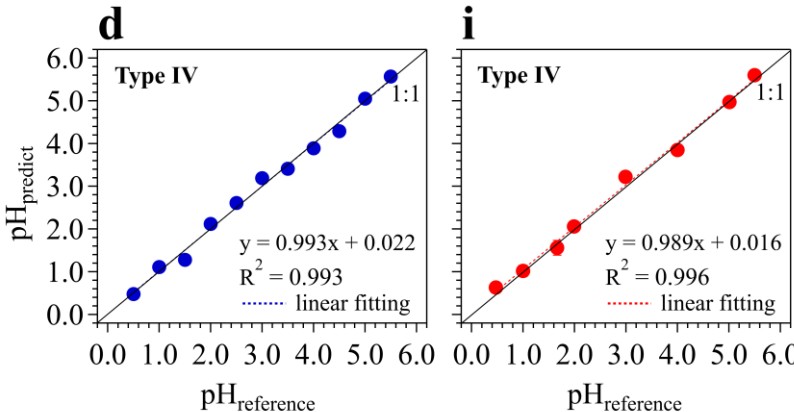

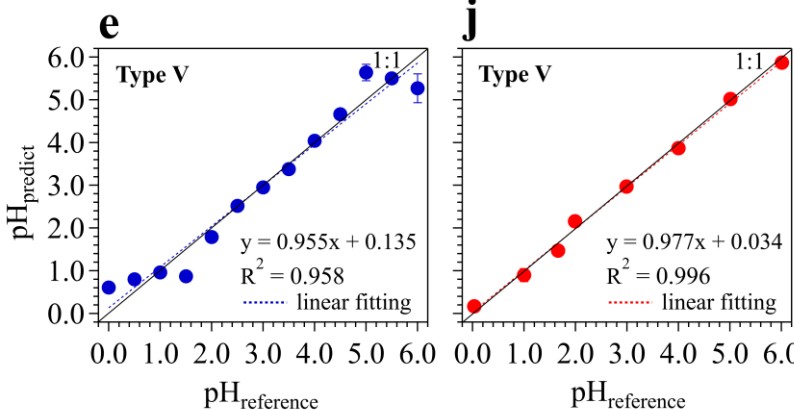




**Figure 2:** Predicted pH ($pH_{predict}$) using our RGB model versus the reference pH shown on the color chart and the pH-meter-
probed-pH of the buffer samples (all denoted as $pH_{reference}$) respectively, for the five different pH papers: (a) and (f) Type I: 0
– 2.5, (b) and (g) Type II: 2.5 – 4.5, (c) and (h)  Type III: 4.0 – 7.0, (d) and (i)  Type IV: 0.5 – 5.5 and (e) and (j)  Type V: 0 –
6.0. Blue symbols denote the established relationship based on color charts only. Red symbols represent the results for 2 μL
of buffer droplets on pH papers. Both vertical and horizontal error bars represent the standard deviation of five to six replicate
experiments. Note that the error bars in most of the panels are smaller than the symbols.

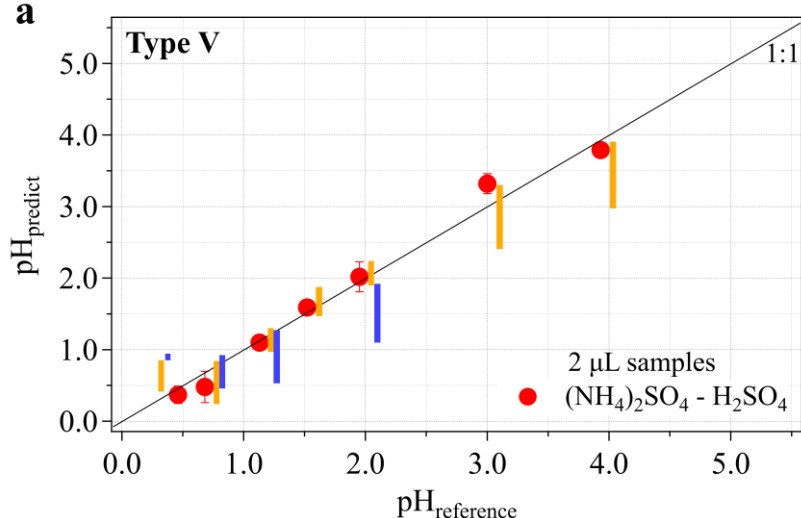


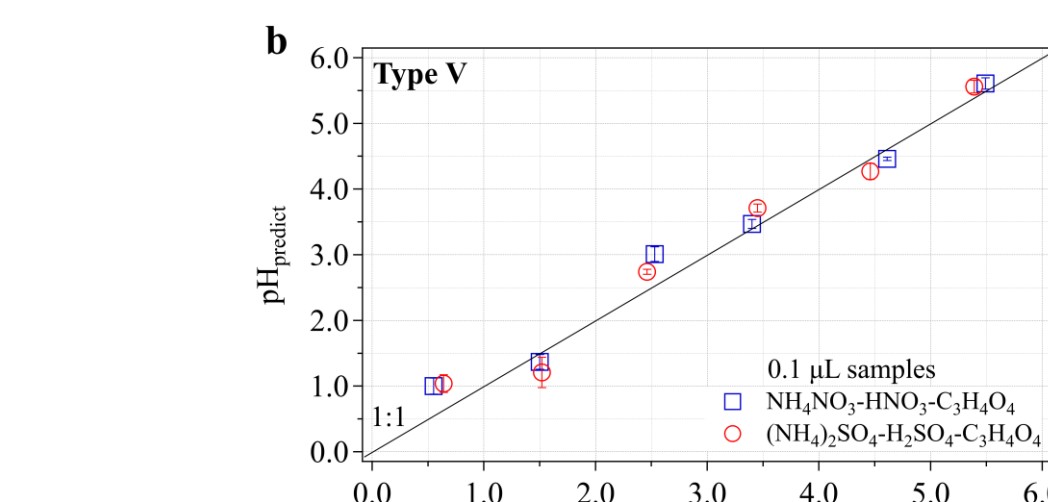



**Figure 3:** pH estimation using the type V pH paper for samples with different volumes: (a) 2 µL and (b) 0.1 µL. pH$_{predict}$ are calculated with the averaged coefficient vector [*a*, *b*, *c*] derived from three to six replicate experiments with the same amounts of standard buffers as of the samples under constant photographing conditions. The error bars represent the standard deviation of three to six replicate experiments. In (a), the heights of the orange and blue bars indicate the reported pH ranges measured with pH papers and Raman spectroscopy respectively, for $(NH_4)_2SO_4$ - $H_2SO_4$ aerosols with particle sizes larger than 2.5 µm in Craig et al. (2018). Each orange or blue bar has the same pH$_{reference}$ as of the red symbol close to it. In (b), for processing the digital images of the 0.1 µL samples, a square with $20 \times 20$ pixels at the center of the samples is cropped for subsequent colorimetric analyses.






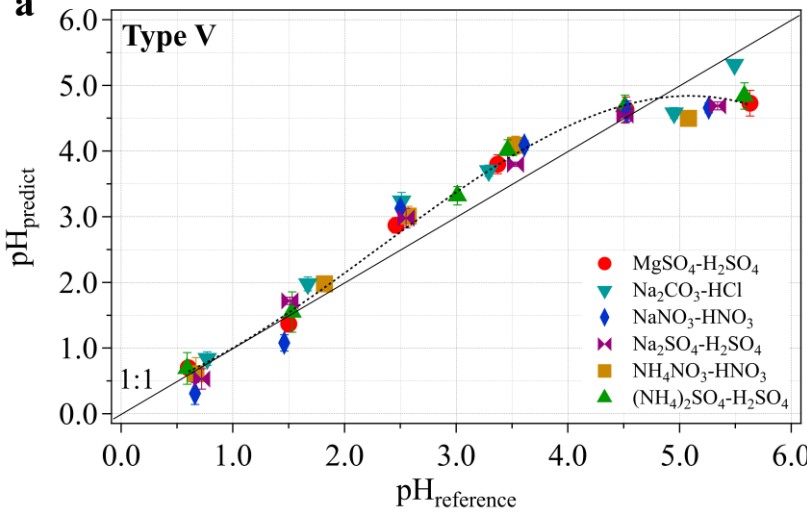


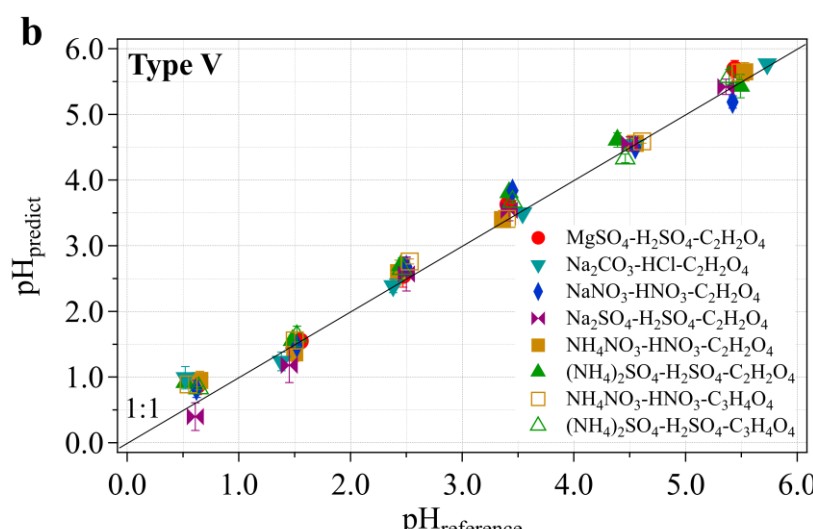


**Figure 4:** pH estimation using the type V pH paper for salt systems with only inorganic acids (a) and both inorganic and organic acids (b). $pH_{predict}$ are calculated with the averaged coefficient vector [$a$, $b$, $c$] derived from three replicate calibration experiments with standard buffers and under constant photographing conditions. The error bars represent the standard deviation of three to four replicate experiments. The dotted line in (a) is used to guide the eye.


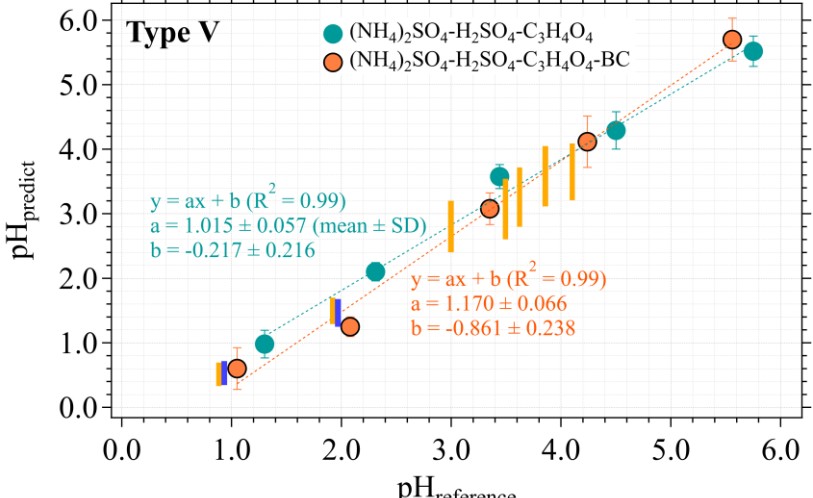

**Figure 5:** pH estimation using the type V pH paper for lab-generated aerosols with or without the co-existence of black carbon (BC). $pH_{predict}$ are calculated with the averaged coefficient vector [a, b, c] derived from five replicate calibration experiments with standard-buffer-generated aerosol samples. The error bars represent the standard deviation of three replicate experiments. The heights of the orange and blue bars indicate the reported pH ranges measured with pH papers and Raman spectroscopy respectively, for $(NH_4)_2SO_4$ - $H_2SO_4$ aerosols with particle sizes in the range of 0.4 - 2.5 μm in Craig et al. (2018). At $pH_{reference}$ < 2.5, each orange or blue bar has the same $pH_{reference}$ as of the orange symbol close to it. Image processing of the collected aerosol samples follows a similar procedure as described in Sect. 2.3.