# Peer review of "Multifactor colorimetric analysis on pH-indicator papers: an"

_Atmospheric Measurement Techniques, 2019_

## Referee Comment (RC1) · Anonymous Referee #1 · 11 Nov 2019

This work proposed a new model to establish the correlation between the color of samples on pH-indicator papers and their measured pH. This model was based on RGB analysis of the images of samples. Good agreement between the model-predicted pH and reference pH for pH paper color charts as well as standard buffers were observed for all the tested types of pH papers. The minimum liquid sample mass/volume needed for the type V pH paper is identified as $\sim$ 180 $\mu$g/0.1 $\mu$L. Aerosol pH measurement is important for understanding the properties of aerosols. This work provided an improved model to do this. It is of sicientific interest, and conclusions are supported by the data. However, two major concerns are: 1. The real application of this method is not performed. In real application, there will be many solid particles as well, which also

have color (for example, black carbon) and may intefere with the measurement. This needs to be clarified. 2. In this work, a mobile phone camera is used to capture the color, which limits the the minimum liquid sample mass/volume needed for the type V pH paper. Even $0.1\mu$L is still too much. The measured pH value will be a collective result of many aerosols. To get pH information of one individual aerosol is more interesting. Using an optical microscope may be more accurate and can further reduce the limits.

---

## Referee Comment (RC2) · Anonymous Referee #2 · 5 Feb 2020

The authors build on the work of Craig et al using particles collected on pH indicator paper as a way to quantify actual ambient particle pH. The focus of this work is on the analysis of the color of the pH paper, not if the overall concept is of collecting particles on pH paper to determine pH is feasible. For example, there is no assessment in this work of the Craig et al method. The authors point out the challenging issues with determining particle pH; the small amounts of water that one has to work with, and that the liquid concentrations of the all the ions in the particle must be precisely maintained throughout the whole period spanning particle collection to pH measurement. The latter means no gradients in water or ions on the pH paper, no changes in T, RH, concentrations of semivolatile gases (ie, $HNO_3$, HCl, organic acids) from ambient to

the environment of the pH indicator strip during sampling or during pH analysis based on color. Achieving these criteria seems extremely difficult and at this point has never been proven to be accurate for fine particles, as far as I can tell by looking at the Craig et al results for PM2.5. Furthermore, the authors use a highly simplified way of applying the surrogate for particle water to the pH paper and justify the amount of water in the tests with unrealistic (possibly meaningless) calculations. So the question is, is this research worthy of publication if the fundamental method on which it is based is possibly flawed (or impossible to achieve in practice), even if the colorimetric analysis, the main focus of this paper is reasonable? My suggestion is that the paper not be published until the authors 1) provide a detailed assessment of the results of Craig et al. discussing if it is viable and if so under what conditions (example, mainly just Dp>2.5 um, collected cloud water, etc), and 2) show that they can actually use this approach to accurately measure pH of real ambient particles, which is the objective of this research, otherwise there is an implied indorsement of this approach. 3) Assess the overall pH uncertainty of an actual process/instrument that could utilize their color analysis for different types of aerosols (ie, fine, coarse, cloud/fog water) under different ambient conditions (concentrations, RH). Alternatively, the authors could change the focus of the paper to simply one on using a smart phone to assess the color of pH paper, which may be of value when the pH indicator paper is used in the method it was intended for, measuring pH of bulk solutions.

More details are provided below:

In the Title or Abstract please specify what size of particles the method will be used to determine particle pH.

This paper does not address the fundamental question if the overall concept of Craig et al is practical or valid, nor does it critically assess the Craig results. The authors simply accept the method. The Craig et al data show that there is significant difficulty with the method for particles smaller than 2.5 um. The authors should first assess if the approach is feasible (see discussion on this below) before claiming to have developed

a method for measuring particle pH.

pH paper is used to measure pH of a large bulk solution, particles collected on a spot are not equivalent to this process. Please discuss the issues for accurate pH measurement with indicator paper due to these differences. This could include, evaporation of water from the paper, changes in ion activities when added to the paper and adsorbed by the paper (ie, wicked away from the original spot of application). Some of these points are discussed later in the paper, it would be good if this was discussed first.

The approach to test the method is to start with a bulk solution of known pH and then pipet a small amount (2 $\mu$L) on to pH paper in a lab with no environmental controls (T, RH =?) to simulate particle collection, then to measure pH by color analyses. The authors state this amount of liquid could be obtained for the following situation (quoted from the paper lines 139-143):

This adopted small volume (2 $\mu$L) was based on calculation of the available amounts of liquid aerosols for aerosol sampling under a typically polluted conditions (with PM2.5 mass concentration around 100 $\mu$g m-3) with high relative humidity ($\geq$ 80%), and assuming a sampling flow rate of several hundred liter per minute (e.g., can be achieved by a Tisch Environmental PM2.5 high volume air sampler, see https://tisch-env.com/high-volume-air-sampler/pm2.5) and a sampling time of a few (2 - 4) hours.   Please comment on how typical these conditions are.

The discussion from the paper copied above seems to only consider how much particle liquid water is available in theory, not how it will be actually collected and how this will compare to the pipetting of 2uL. Since the measurement is based on liquid water on the filter, one needs to know the size (surface area) of the filter (ie, how much area the collected water will be spread over). Is the liquid water spread evenly across (and possibly within the filter)? How would one maintain identical conditions on the filter as in ambient air during the sample and analysis time, which is critical to an accurate pH measurement? Can pH paper be used as a particle filter, if not how does one filter the

particles and then use the pH paper, ie do the authors envision collecting the water on the filter and then use the pH paper to measure pH of that water, is this possible?

How big of a spot (ie, diameter) on the pH paper is the liquid when 2 $\mu$L are pipetted onto the paper? Does spot size matter? Is it practically possible to collect particles in that spot size so as to mimic the experiments performed here? For example, given the conditions above and the sample flow rate is there a devise that can achieve this. The high volume sampler suggested collects particle over a large area (ie, a filter), so as noted above the question is, will this work as a method to collect the particles? Presumably to collect a spot of particle water, one would have to do this with an impactor. To not change the concentration of the ions that exists in the particle, the wicking away of the water on the collection paper would have to be limited, or at least the ions wicked away at the same rate as water. With an impactor the drop spreading would be enhanced by the air jet moving along the surface of the collection paper. Furthermore, how will pressure drops across the impactor or filter affect the determined pH (ie, loss of liquid water, etc). One should also consider other possible sampling issues that could alter pH from ambient?

It is noted in Section 4.2 that the pH could be determined by this method for much smaller volumes, requiring a lower sampling time and/or lower sampler flow rate. However, as notes, evaporation then becomes important and the measurements must be made rapidly, ie < 3 sec (for the conditions in their lab, ie what was the ambient RH in the lab). Again, is it really possible (practical) to actually use such small liquid samples as described in this approach?

Given all the possible issues with accurate pH determination of fine PM with this method, combined with the uncertainty in interpreting the pH indicator color (line 235-236), and that it is noted by the authors that uncertainty in pH can have huge impacts on pH dependent multiphase chemical processes (lines 86-87), is this method really a reasonable way to determine particle pH? The authors need to supply an actual estimate in the uncertainty in the pH determined by this method so that one can assess

the impact it will have when these pH values are used.

---

## Editor Comment (EC1) · Mingjin Tang (Editor) · 5 Mar 2020

Review of "Multifactor colorimetric 1 analysis on pH-indicator papers: an optimized approach for direct determination of ambient aerosol pH"

This manuscript describes an improvement upon a recently published method (Craig et al. 2018) using image processing of colorimetric indicator paper to analyze the pH of atmosphere particles. The work is thorough and worthy of publication. There a few points I would suggest addressing and one major weakness to the manuscript. Overall, this method is an important step forward for determination of aerosol pH.

The largest concern with the manuscript is that it all of the analysis is with pipetted solutions and not with actual lab-generated or ambient aerosol as far as I can tell. Both Craig et al. 2018 and Coddens et al. 2019 from the Grassian laboratory looked at suspended aqueous aerosol that were then impacted onto colorimetric indicator paper. This led to some unique results (e.g. size dependence of pH), which make it not surprising that the 0.1 microlitre samples herein rapidly changed after pipetting on the paper. Even running just a few aerosolized samples to verify the selection of the specified pH paper would greatly strengthen the manuscript.

A minor is the justification of using 2 microlitres samples overall based on a high volume sampler pulling hundreds of lpm for a couple of hours. With that kind of flow rate and timing, a sample is unlikely to retain this amount of water due to drying and, at a minimum would be vastly altered at the end of sampling versus what was initially collected. Losses of semi-volatile inorganic (e.g. ammonium/ammonia) and organic (e.g. carboxylic acids like acetic acid) species would be expected in that sampling setup.

For Figure 2 it would be helpful to include both x- and y-error bars on the points, with x representing the uncertainty in the predicted pH and y the uncertainty in the pH probe/buffer measurements. This would help to know if the uncertainties include the regression line for the points that do not fall exactly on it.

This is a small point, but the term "outlier" is probably not the best for the point on Figure S4. If it is reproducible to the extent described it is by definition not an outlier. I think "anomalous" might be a better term, as this point would not be thrown out by the traditional Grubbs test of an outlier or other outlier tests.

The last point would be to that though the mention "anti-interference" it would be useful for the authors to see if their RGB method would work with brown carbon or black carbon samples (or some other chromophoric aerosol) that also contain secondary species and water.

---

## Author Comment (AC1) · 30 Aug 2020

**Response to Anonymous Referee #1**

We thank the reviewer for the constructive suggestions/comments. Below we provide a point-by-point response to individual comment (Reviewer comments and suggestions are in italics, responses and revisions are in plain font; revised parts in responses are marked with red color; page numbers refer to the modified AMTD version).

**Comments and suggestions:**

*Overall Comments. This work proposed a new model to establish the correlation between the color of samples on pH-indicator papers and their measured pH. This model was based on RGB analysis of the images of samples. Good agreement between the model-predicted pH and reference pH for pH paper color charts as well as standard buffers were observed for all the tested types of pH papers. The minimum liquid sample mass/volume needed for the type V pH paper is identified as ~ 180 µg/0.1 µL. Aerosol pH measurement is important for understanding the properties of aerosols. This work provided an improved model to do this. It is of scientific interest, and conclusions are supported by the data.*

**Responses and Revisions:**

Thanks for the positive comments and feedback from the reviewer.

**Comments and suggestions:**

*Specific Comments. However, two major concerns are: 1. The real application of this method is not performed. In real application, there will be many solid particles as well, which also have color (for example, black carbon) and may interfere with the measurement. This needs to be clarified.*

**Responses and Revisions:**

Good suggestion.

To further test the feasibility of our method for real aerosol samples, we generated aerosol particles under laboratory conditions and collected them on pH papers by using two custom-made impactors which were connected in series during the aerosol sampling. In addition, the potential interference of black carbon (BC) on aerosol pH prediction was also examined. Generally, we could reasonably

predict the pH of lab-generated aerosols based on the colorimetric analysis method proposed in the manuscript. The results also confirm the technical feasibility of collecting aerosol particles on pH papers through impactors. Moreover, the potential interference of BC on aerosol pH prediction was proved to be non-significant when we adopted a BC concentration representative of ambient BC levels. Details can be found in the revised part as follows:

[revised manuscript text omitted]

**Comments and suggestions:**

*Specific Comments. 2. In this work, a mobile phone camera is used to capture the color, which limits the minimum liquid sample mass/volume needed for the type V pH paper. Even 0.1 µL is still too much. The measured pH value will be a collective result of many aerosols. To get pH information of one individual aerosol is more interesting. Using an optical microscope may be more accurate and can further reduce the limits.*

**Responses and Revisions:**

Thanks for the comments.

To the best of our knowledge, currently available techniques for measuring the pH of one individual aerosol include Raman microspectroscopy (Rindelaub et al., 2016; Wei et al., 2018; Lei et al., 2020) and aerosol optical tweezers (Boyer et al., 2020), which are all deployed in laboratories. Generally speaking, these techniques are mainly used for probing the pH of aerosols/droplets with larger sizes than we have in this study (0.4 – 2.2 µm in diameter). For example, the investigated aerosols had a diameter size range of 6 – 10 µm for the aerosol optical

tweezer technique used by Boyer et al. (2020). And the Raman microspectroscopy was used to measure the pH of droplets with a diameter range of 10 – 30 μm and 13.3 – 25.7 μm in the work of Rindelaub et al. (2016) and Wei et al. (2018), respectively. Even though in the most recent study Lei et al. (2020) examined the pH of submicron aerosols, their experiments were performed through using Raman spectroscopy coupled to atomic force microscopy under well-controlled laboratory conditions. For our pH paper method and the way of aerosol sampling, collecting a large amount of aerosols on the pH paper seems to be the only available and feasible approach for pH estimation of submicron particles. More importantly, our method can be easily used for ambient aerosol pH prediction and therefore our way of exploring the aerosol pH would be more representative of ambient cases.

---

## Author Comment (AC2) · 30 Aug 2020

**Response to Anonymous Referee #2**

We thank the reviewer for the constructive suggestions/comments. Below we provide a point-by-point response to individual comment (Reviewer comments and suggestions are in italics, responses and revisions are in plain font; revised parts in responses are marked with red color; page numbers refer to the modified AMTD version).

**Comments and suggestions:**

*Overall Comments. The authors build on the work of Craig et al using particles collected on pH indicator paper as a way to quantify actual ambient particle pH. The focus of this work is on the analysis of the color of the pH paper, not if the overall concept is of collecting particles on pH paper to determine pH is feasible. For example, there is no assessment in this work of the Craig et al method.*

**Responses and Revisions:**

Thanks for the comments.

The colorimetric analysis on pH papers is based on a prerequisite that aerosol samples can be collected on pH papers. In the work of Craig et al. (2018), the authors showed that aerosols generated in the lab as well as from ambient air were impacted onto pH papers by using a microanalysis particle sampler (MPS-3, California Measurements, Inc.). The MPS-3 had three stages with aerodynamic diameter cutoff sizes ($d_{50}$) of 2.5-5.0, 0.4-2.5, and < 0.4 μm for stages 1, 2 and 3, respectively. To collect lab-generated aerosols, the authors let the originally undried particles impact on the pH papers, to ensure the collected aerosol particles were aqueous (Craig et al., 2018). For ambient aerosol sampling, ambient aerosol samples were collected for ~1-2 hours with an ambient RH range of 60% - 80% (Craig et al., 2018). All these results shown by Craig et al. (2018) have demonstrated the potential of the pH paper method for aerosol collection and pH prediction. On the basis of Craig et al' work, we further proposed the optimized RGB model for colorimetric analysis on pH paper colors. Moreover, we further tested the feasibility of collecting lab-generated aerosols on pH papers. The related results have been added into the revised manuscript.

*Comments and suggestions:*

*Overall Comments. The authors point out the challenging issues with determining particle pH; the small amounts of water that one has to work with, and that the liquid concentrations of the all the ions in the particle must be precisely maintained throughout the whole period spanning particle collection to pH measurement. The latter means no gradients in water or ions on the pH paper, no changes in T, RH, concentrations of semivolatile gases (ie, $HNO_3$, HCl, organic acids) from ambient to the environment of the pH indicator strip during sampling or during pH analysis based on color. Achieving these criteria seems extremely difficult and at this point has never been proven to be accurate for fine particles, as far as I can tell by looking at the Craig et al results for PM2.5.*

**Responses and Revisions:**

Thanks for the comments.

Indeed, to determine the pH of aerosol particles one will face many challenging issues. One of the most critical issues is the aerosol pH is dynamically changing due to the variation of surrounding environment (e.g., temperature, relative humidity, concentration of different gas species). From ambient aerosol sampling to a new environment where the pH paper colors/images are captured, the original aerosol pH may already be altered. To avoid this problem, we have designed two impactors. One piece of pH paper could be fixed on the impactor bottom plate and one camera was installed on the top of the impactor. During a sampling process, aerosols with certain sizes could impact on the pH paper and the induced color change could be monitored and captured by the camera. With this impactor setup, we could check the change of the pH paper color at any sampling time without interrupting the sampling. Moreover, we would expect a small difference between the surrounding environment of aerosols inside the impactor and that outside the impactor (i.e., ambient cases). Therefore, in this way the estimated aerosol pH should be representative of the aerosol acidity under ambient cases. More details regarding aerosol collection with the impactor and the related experiment results can be found in the revised manuscript.

*Comments and suggestions:*

*Overall Comments. Furthermore, the authors use a highly simplified way of applying the surrogate for particle water to the pH paper and justify the amount of water in the tests with unrealistic (possibly meaningless) calculations.*

**Responses and Revisions:**

Thanks for the comments.

The aerosol surrogates reflected the main/common components existed in ambient aerosols, and in this study, they were used to check the performance and anti-interference capacity of different types of pH papers. Thus, using these surrogates could help us quickly find some types of pH papers which are more suitable for ambient aerosol pH measurements. The conversion from the tested volume (i.e., 0.1 μL) of aerosol sample solutions to the needed minimum amount for pH color change could provide us a general estimation on the lower limit of the needed volume/mass for ambient aerosol samplings. In the revised manuscript, we further estimated the sampling time needed for ambient aerosols, based on the sampling time of collecting lab-generated aerosols on pH papers through custom-made impactors. This estimation would make more sense than just based on the tested volume of aerosol surrogates.

*Comments and suggestions:*

*Overall Comments. So the question is, is this research worthy of publication if the fundamental method on which it is based is possibly flawed (or impossible to achieve in practice), even if the colorimetric analysis, the main focus of this paper is reasonable?*

**Responses and Revisions:**

Thanks for the comments.

As stated above, the colorimetric analysis on pH papers (the main focus of our study) is based on a prerequisite that aerosol samples can be collected on pH papers. In the work of Craig et al. (2018), the authors have shown that aerosols generated in the lab as well as from ambient air were impacted onto pH papers by using a microanalysis particle sampler (MPS-3, California Measurements, Inc.). In our revised manuscript, we further added one section (Sect. 4.2) to demonstrate that collection of aerosols on pH papers is feasible.

*Comments and suggestions:*

*Overall Comments. My suggestion is that the paper not be published until the authors 1) provide a detailed assessment of the results of Craig et al. discussing if it is viable and if so under what conditions (example, mainly just Dp>2.5 um, collected cloud water, etc), and 2) show that they can actually use this approach to accurately measure pH of real ambient particles, which is the*

*objective of this research, otherwise there is an implied indorsement of this approach. 3) Assess the overall pH uncertainty of an actual process/instrument that could utilize their color analysis for different types of aerosols (ie, fine, coarse, cloud/fog water) under different ambient conditions (concentrations, RH). Alternatively, the authors could change the focus of the paper to simply one on using a smart phone to assess the color of pH paper, which may be of value when the pH indicator paper is used in the method it was intended for, measuring pH of bulk solutions.*

**Responses and Revisions:**

Thanks for the suggestions.

Regarding the first suggestion, we have added the results of Craig et al. into the "Introduction" part, and the detailed discussion based on their results and our results has also been added into the Section of 4.2, in the revised manuscript.

For the second suggestion, we further performed experiments to confirm that the lab-generated aerosols could be collected onto pH papers by using a custom-made impactor. Moreover, our results also demonstrate that, with our proposed RGB model the pH paper method can be used to predict aerosol pH with a high accuracy. The application of the impactor setup as well as the pH paper method in real ambient cases will be explored in our future work. Details can be found in Section 2.2 and 4.2, in the revised manuscript.

For the third suggestion, we designed an impactor and applied it to collect aerosol particles on pH papers under lab conditions. The overall uncertainty of the predicted aerosol pH was ≤ 0.5 unit. The aerosol sampling and pH prediction were carried out under a RH of ~ 90% and with a total aerosol mass concentration of ~ 800 μg m$^{-3}$. Details can be found in Section 2.2 in the revised manuscript. The application of the impactor setup as well as the pH paper method under different ambient conditions will be explored in our future work.

**Comments and suggestions:**

*Specific Comments. In the Title or Abstract please specify what size of particles the method will be used to determine particle pH.*

**Responses and Revisions:**

Thanks for the comments.

The particle size range has been specified in the abstract. As shown below:

"… Custom-made impactors are used to collect lab-generated aerosols on this type of pH paper. Preliminary tests show that, with a collected particle size range of ~ 0.4 – 2.2 µm, the pH paper method can be used to predict aerosol pH with an overall uncertainty ≤ 0.5 unit. Based on laboratory tests, a relatively short sampling time (~ 1 to 4 hours) is speculated for pH prediction of ambient aerosols. More importantly, our design of the impactors minimizes potential influences of changed environmental conditions during pH paper photographing processes on the predicted aerosol pH. …"

**Comments and suggestions:**

*Specific Comments. This paper does not address the fundamental question if the overall concept of Craig et al is practical or valid, nor does it critically assess the Craig results. The authors simply accept the method. The Craig et al data show that there is significant difficulty with the method for particles smaller than 2.5 µm. The authors should first assess if the approach is feasible (see discussion on this below) before claiming to have developed a method for measuring particle pH.*

**Responses and Revisions:**

Thanks for the comments.

We have added a detailed discussion based on Craig et al.' results and our results into the Section of 4.2, in the revised manuscript. It is not clear why the reviewer considered the overall concept of Craig et al. to be flawed. If the reviewer' concern came from "*Craig et al data show that there is significant difficulty with the method for particles smaller than 2.5 µm*", we hope that through the following detailed illustration we can justify the validity of the pH paper method for aerosol pH estimation.

In the work of Craig et al. (2018), the pH-paper-derived results of aerosol acidity increasing with decreasing particle size were further confirmed by comparisons with direct measurements of individual aerosol particle pH via a Raman microspectroscopy technique. During aerosol transportation from the aerosol generator to the pH paper, the generated aerosols may have undergone water exchange with the surrounding air and the chemical equilibrium for aqueous reactions inside aerosols may be shifted due to the pH effect. These may lead to water loss and/or gas partitioning (such as ammonia) between particles and surrounding gases. These changes are more likely to occur for smaller particles, due to their higher surface-area-to-volume ratios. This means that before impacting on pH papers, the aerosols have already had a different pH from its original state. That is why the pH paper will show a different aerosol pH from that of bulk

solutions. For lab-generated liquid-phase aerosols, the potential water exchange between aerosols and surrounding gases can be minimized by using a gas flow with a similar RH to that of the generated aerosol flow (we did this for our lab experiments, more details can be found in Section 2.2). However, the gas partitioning caused by the intrinsic chemical equilibrium in the particle phase cannot be avoided under lab conditions. Therefore, the results reported by Craig et al. doesn't mean the pH paper method is infeasible. For ambient aerosols, their pH is dynamically changing due to the variation of surrounding environment (e.g., temperature, relative humidity, concentration of different gas species). Considering this critical issue, we designed one type of impactor. One piece of pH paper could be fixed on the impactor bottom plate and one camera was installed on the top of the impactor. During a sampling process, aerosols with certain sizes could impact on the pH paper and the induced color change could be monitored and captured by the camera. With this setup, we could check the pH paper color at any sampling time without interrupting the sampling. Moreover, we would expect a small difference between the surrounding environment of aerosols inside the impactor and that outside the impactor (i.e., ambient cases). Therefore, in this way the estimated aerosol pH should be representative of the aerosol acidity under ambient cases. More details regarding aerosol collection with the impactor and the related experiment results can be found in Section 2.2 and 4.2. As shown below.

[revised manuscript text omitted]

**Comments and suggestions:**

*Specific Comments. pH paper is used to measure pH of a large bulk solution, particles collected on a spot are not equivalent to this process. Please discuss the issues for accurate pH measurement*

*with indicator paper due to these differences. This could include, evaporation of water from the paper, changes in ion activities when added to the paper and adsorbed by the paper (ie, wicked away from the original spot of application). Some of these points are discussed later in the paper, it would be good if this was discussed first.*

**Responses and Revisions:**

Thanks for the comments.

As the reviewer pointed out, the potential influences from surroundings and the pH paper itself can become more critical for pH measurements of particles than for bulk solutions. Thus, the experiment conditions need to be well-controlled and the way of aerosol sampling needs to be carefully designed during particle pH measurements in order to minimize these effects. Based on our experimental results (Sect. 4.2), we have demonstrated that these potential influences were small for our case.

Moreover, we have added one discussion part into the "Introduction" section. As shown below.

"… Additionally, due to the small area and various shape of different types of pH papers, collection of aerosols on these materials is quite distinct from that on commonly used filters. The collected particles may induce a color change only on a small spot (Craig et al., 2018), differing from the color variation on a much larger scale caused by bulk solutions. Moreover, the environment under which aerosols are collected can indirectly affect the measured aerosol pH: In an environment different from that the aerosols were originally in, evaporation/condensation of water on pH papers might happen, which may further lead to changes in ion activities and/or water dispersion/homogeneity on pH papers. Thus, to have accurate aerosol pH measurements, special techniques/instruments need to be developed for effective aerosol collection and pH paper color recognition, and meanwhile careful design should be made to avoid potential impacts of varied environmental factors on the predicted aerosol pH."

**Comments and suggestions:**

*Specific Comments. The approach to test the method is to start with a bulk solution of known pH and then pipet a small amount (2 µL) on to pH paper in a lab with no environmental controls (T, RH =?) to simulate particle collection, then to measure pH by color analyses. The authors state this amount of liquid could be obtained for the following situation (quoted from the paper lines 139-143):*

*This adopted small volume (2 µL) was based on calculation of the available amounts of liquid aerosols for aerosol sampling under a typically polluted conditions (with PM2.5 mass concentration around 100 µg m$^{-3}$) with high relative humidity ($\geq$ 80%), and assuming a sampling*

*flow rate of several hundred liter per minute (e.g., can be achieved by a Tisch Environmental PM2.5 high volume air sampler, see https://tischenv.com/high-volume-air-sampler/pm2.5) and a sampling time of a few (2 - 4) hours.*

*Please comment on how typical these conditions are.*

**Responses and Revisions:**

Thanks for the suggestion.

We have added some comments on the typical conditions which were used for our estimation. As shown below.

"… This adopted small volume (2 µL) was based on a general estimation of the available amounts of liquid aerosols for aerosol sampling under a typically polluted conditions (with $PM_{2.5}$ mass concentration around 100 µg m$^{-3}$) with high RH (60% – 80%), and assuming an aerosol collection efficiency of 50% and a sampling flow rate of several hundred liter per minute (e.g., can be achieved by a Tisch Environmental $PM_{2.5}$ high volume air sampler, see https://tisch-env.com/high-volume-air-sampler/pm2.5) with a sampling time of a few (2 - 4) hours. Here, the used $PM_{2.5}$ mass concentration and RH refer to the conditions during haze events which are frequently occurring in China. For example, during the most severe haze episodes in January 2013, monthly averaged $PM_{2.5}$ concentration in Beijing reached 121 µg m$^{-3}$ and the RH was constantly at a level of 60% – 80% (Zheng et al., 2015). Even the air quality in China has significantly improved in recent years, the number of days with moderate haze (with daily mean $PM_{2.5}$ concentration in the range of 100 – 200 µg m$^{-3}$) in the North China Plain shows on obviously decreasing trend from 2004 to 2018 with an average of 113 d (Zhang et al., 2020). Note that, we further estimated the minimum sample volume and mass needed to generate a measurable color change on the suggested pH paper. …"

**Comments and suggestions:**

*Specific Comments. The discussion from the paper copied above seems to only consider how much particle liquid water is available in theory, not how it will be actually collected and how this will compare to the pipetting of 2µL. Since the measurement is based on liquid water on the filter, one needs to know the size (surface area) of the filter (ie, how much area the collected water will be spread over). Is the liquid water spread evenly across (and possibly within the filter)? How would one maintain identical conditions on the filter as in ambient air during the sample and analysis time, which is critical to an accurate pH measurement? Can pH paper be used as a particle filter, if not how does one filter the particles and then use the pH paper, ie do the authors envision collecting the water on the filter and then use the pH paper to measure pH of that water, is this possible?*

**Responses and Revisions:**

Thanks for the comments.

Yes, the reviewer is right. During calculation, we shouldn't only consider the amount of available aerosol liquid water. The volume/mass of collected aerosols also depends on how the aerosols are collected on pH papers (i.e., the aerosol collection efficiency). To collect aerosols on pH papers, we further built two impactors. One piece of pH paper ($5 \times 5$ mm) could be fixed on the impactor bottom plate and one camera was installed on the top of the impactor. During a sampling process, aerosols with certain sizes could impact on the pH paper and the induced color change could be captured by the camera. Here the pH paper worked as an inertial impaction filter because aerosols were impacted onto the pH paper due to a sudden change of the aerosol flow direction inside the impactor. Based on our preliminary tests, the pH paper showed a collection efficiency of ~ 55% – 99% for aerosols in the size range of 0.4 – 2.2 µm (i.e., 55% for 0.4 µm aerosols and 99% for 2.2 µm aerosols). This aerosol collection efficiency was generally comparable to that of other types of inertial impaction filters, which was in a range of 17% – 100% for $PM_{2.5}$ (Zhang et al., 2018).

Moreover, we would also expect that a temperature and humidity control system could help to reduce the difference between surrounding environment of aerosols inside the impactor and that outside the impactor (i.e., ambient cases). Based on our preliminary tests by using lab-generated aerosols, the impactor setup has been demonstrated to work well for aerosol collection and pH prediction, and therefore has a great potential for future application under ambient conditions. During our lab tests, an optimal aerosol sampling time was identified as 30 min and the generated aerosol concentrations were ~ 800 µg m$^{-3}$ (measured under RH = 14%), the collected aerosols could cause a generally uniform color change on the entire area of the pH paper. And this uniform color change indicated that the liquid aerosols had spread evenly across the pH paper. Thus, the whole pH paper area was used for subsequent image processing to get their RGB values. According to the measurement results under lab conditions, we further estimated a sampling time range of ~ 1 – 4 hours (with a sampling flow range of ~ 30 – 120 L min$^{-1}$) for ambient aerosols with a $PM_{2.5}$ concentration of ~ 100 µg m$^{-3}$. Based on APS measurements, the mass collection efficiency of the impactor was estimated as 70% – 90%. With these sampling parameters, we further identified a collected aerosol mass (on the pH paper) of 480 – 617 µg under RH = 14% and of 4.8 – 6.2 mg under RH = 90%. And the aerosol mass under the high RH corresponded to a volume of 2.7 – 3.2

μL, assuming an effective density of 1.8 g cm$^{-3}$ for ambient aerosols (Sarangi et al., 2016; Geller et al., 2006).   Therefore, the adopted volume of 2 μL for our pH paper tests generally agreed with the amount speculated based on experiments.   We have adjusted the related calculation (for 2 μL) part, as shown below:

"… This adopted small volume (2 μL) was based on a general estimation of the available amounts of liquid aerosols for aerosol sampling under a typically polluted conditions (with PM$_{2.5}$ mass concentration around 100 μg m$^{-3}$) with high RH (60% – 80%), and assuming an aerosol collection efficiency of 50% and a sampling flow rate of several hundred liter per minute (e.g., can be achieved by a Tisch Environmental PM$_{2.5}$ high volume air sampler, see https://tisch-env.com/high-volume-air-sampler/pm2.5) with a sampling time of a few (2 - 4) hours. …"

**Comments and suggestions:**

*Specific Comments. How big of a spot (ie, diameter) on the pH paper is the liquid when 2 μL are pipetted onto the paper? Does spot size matter? Is it practically possible to collect particles in that spot size so as to mimic the experiments performed here? For example, given the conditions above and the sample flow rate is there a devise that can achieve this. The high volume sampler suggested collects particle over a large area (ie, a filter), so as noted above the question is, will this work as a method to collect the particles? Presumably to collect a spot of particle water, one would have to do this with an impactor. To not change the concentration of the ions that exists in the particle, the wicking away of the water on the collection paper would have to be limited, or at least the ions wicked away at the same rate as water. With an impactor the drop spreading would be enhanced by the air jet moving along the surface of the collection paper. Furthermore, how will pressure drops across the impactor or filter affect the determined pH (ie, loss of liquid water, etc). One should also consider other possible sampling issues that could alter pH from ambient?*

**Responses and Revisions:**

Thanks for the comments.

When 2 μL of samples are pipetted on the type V pH paper, the generated spot has a diameter ~ 6 mm. For 0.1 μL samples, the spot has a diameter ~ 1 mm. The color on the spot is generally even to the eye. As can be seen in Fig. 3b in the manuscript, even with 0.1 μL samples the pH paper can perform well for pH prediction, suggesting that the size doesn't matter. As mentioned above, we further built two impactors to collect lab-generated aerosols onto one piece of pH paper (5 × 5 mm), which was fixed on the impactor bottom plate. And after 30 min of sampling, the aerosolinduced color change could be observed on the whole pH paper area. This pH paper size was comparable to the spot generated by 2 µL samples. For our preliminary tests on the impactors, a sampling flow of 28.6 L min$^{-1}$ was used. With two impactors connected in series, a total pressure drop of ~ 57 mbar was detected. This pressure drop had no big effect on the predicted pH, which can be seen from the measured aerosol pH shown in Fig. 5.

*Comments and suggestions:*

*Specific Comments. It is noted in Section 4.2 that the pH could be determined by this method for much smaller volumes, requiring a lower sampling time and/or lower sampler flow rate. However, as notes, evaporation then becomes important and the measurements must be made rapidly, ie < 3 sec (for the conditions in their lab, ie what was the ambient RH in the lab). Again, is it really possible (practical) to actually use such small liquid samples as described in this approach?*

**Responses and Revisions:**

Thanks for the comments.

As shown in the manuscript, with such a small sample volume/mass (0.1 µL/180 µg) and an induced color change on a spot with a diameter of ~ 1 mm, the pH paper can still work well for pH prediction (the uncertainty is ≤ 0.5 unit). This could provide us a general estimation on the lower limit of the needed volume/mass of collected ambient aerosols, which can further guide us to search for new techniques/instruments for aerosol collection as well as color recognition. Based on these tests, we also get to know that with a small sample volume/mass on the pH paper, the pH of collected samples are more likely to be affected by the environment where the images of pH paper colors are captured. To minimize this effect, we further built two impactors. One piece of pH paper (5 × 5 mm) could be fixed on the impactor bottom plate and one camera was installed on the top of the impactor. During a sampling process, aerosols with certain sizes could impact on the pH paper and the induced color change could be captured by the camera. With this impactor setup, we were able to check the pH paper color at any sampling time without interrupting the sampling. Moreover, we would also expect a small difference between the surrounding environment of aerosols inside the impactor and that outside the impactor (i.e., ambient cases). Therefore, for future ambient applications the estimated aerosol pH should be representative of the aerosol acidity under ambient cases. Based on our preliminary tests by using lab-generated aerosols, the impactor setup has been demonstrated to work well for aerosol collection and pH

prediction, and therefore has a great potential for future application under ambient conditions. More details can be found in Sect. 2.2 and 4.2, in the revised manuscript.

**Comments and suggestions:**

*Specific Comments. Given all the possible issues with accurate pH determination of fine PM with this method, combined with the uncertainty in interpreting the pH indicator color (line 235-236), and that it is noted by the authors that uncertainty in pH can have huge impacts on pH dependent multiphase chemical processes (lines 86-87), is this method really a reasonable way to determine particle pH? The authors need to supply an actual estimate in the uncertainty in the pH determined by this method so that one can assess the impact it will have when these pH values are used.*

**Responses and Revisions:**

Thanks for the comments.

As pointed out in our manuscript and here again by the reviewer, accurate pH determination is another key factor for selecting different methods for ambient aerosol pH prediction. Throughout our manuscript, we have demonstrated that the pH paper method could work well for aerosol pH prediction (with an uncertainty of $\leq 0.5$ unit) when using our proposed RGB model and impactors. This actual estimate in the uncertainty of the pH determined by the pH paper method has also been added in the abstract section. As shown below.

"…Custom-made impactors are used to collect lab-generated aerosols on this type of pH paper. Preliminary tests show that, with a collected particle size range of $\sim 0.4 – 2.2$ µm, the pH paper method can be used to predict aerosol pH with an overall uncertainty $\leq 0.5$ unit. Based on laboratory tests, a relatively short sampling time ($\sim 1$ to 4 hours) is speculated for pH prediction of ambient aerosols. More importantly, our design of the impactors minimizes potential influences of changed environmental conditions during pH paper photographing processes on the predicted aerosol pH. …"

---

## Author Comment (AC3) · 30 Aug 2020

**Response to Anonymous Referee #3**

We thank the reviewer for the constructive suggestions/comments. Below we provide a point-by-point response to individual comment (Reviewer comments and suggestions are in italics, responses and revisions are in plain font; revised parts in responses are marked with red color; page numbers refer to the modified AMTD version).

**Comments and suggestions:**

*Overall Comments This manuscript describes an improvement upon a recently published method (Craig et al. 2018) using image processing of colorimetric indicator paper to analyze the pH of atmosphere particles. The work is thorough and worthy of publication. There a few points I would suggest addressing and one major weakness to the manuscript. Overall, this method is an important step forward for determination of aerosol pH.*

**Responses and Revisions:**

Thanks for the positive comments from the reviewer.

**Comments and suggestions:**

*Specific Comments. The largest concern with the manuscript is that it all of the analysis is with pipetted solutions and not with actual lab-generated or ambient aerosol as far as I can tell. Both Craig et al. 2018 and Coddens et al. 2019 from the Grassian laboratory looked at suspended aqueous aerosol that were then impacted onto colorimetric indicator paper. This led to some unique results (e.g. size dependence of pH), which make it not surprising that the 0.1 microlitre samples herein rapidly changed after pipetting on the paper. Even running just a few aerosolized samples to verify the selection of the specified pH paper would greatly strengthen the manuscript.*

**Responses and Revisions:**

Thanks for the comments.

We further performed experiments to confirm that the lab-generated aerosols could be collected onto pH papers by using two custom-made impactors. Moreover, our results also demonstrate that, with our proposed RGB model the pH paper method can be used to predict aerosol pH with a high accuracy (with a predicted pH uncertainty $\leq 0.5$ unit). The application of the impactor setup as well as the pH paper method in real ambient cases will be explored in our future work. Details can

be found in Section 2.2 and 4.2, in the revised manuscript. As shown below.

[revised manuscript text omitted]

**Comments and suggestions:**

*Specific Comments. A minor is the justification of using 2 microlitres samples overall based on a high volume sampler pulling hundreds of lpm for a couple of hours. With that kind of flow rate and timing, a sample is unlikely to retain this amount of water due to drying and, at a minimum would be vastly altered at the end of sampling versus what was initially collected. Losses of semi‑volatile inorganic (e.g. ammonium/ammonia) and organic (e.g. carboxylic acids like acetic acid) species would be expected in that sampling setup.*

**Responses and Revisions:**

Thanks for the comments.

As stated above, we further built two impactors. One piece of pH paper (5 × 5 mm) could be fixed on the impactor bottom plate and one camera was installed on the top of the impactor. During a sampling process, aerosols with certain sizes could impact on the pH paper and the induced color change could be captured by the camera. With this impactor setup, we were able to check the pH paper color at any sampling time without interrupting the sampling. Moreover, we would also expect a small difference between the surrounding environment of aerosols inside the impactor and that outside the impactor (i.e., ambient cases).

Therefore, for future ambient applications the estimated aerosol pH should be representative of the aerosol acidity under ambient cases. Based on our preliminary tests by using lab-generated aerosols, the impactor setup has been demonstrated to work well for aerosol collection and pH prediction, and therefore has a great potential for future application under ambient conditions. During our lab tests, an optimal aerosol sampling time was identified as 30 min and the generated aerosol concentrations were ~ 800 µg m$^{-3}$ (measured under RH = 14%), the collected aerosols could cause a color change on the whole pH paper. And the colors on the whole area were even to the eye, indicating that the liquid aerosols had spread evenly across the pH paper. Thus, the whole pH paper area was used for subsequent image processing to get their RGB values. According to the measurement results under lab conditions, we further estimated a sampling time range of ~ 1 – 4 hours for ambient aerosols with a PM$_{2.5}$ concentration of ~ 100 µg m$^{-3}$, when using our impactors (with an estimated aerosol collection efficiency of 50% – 70%, based on APS and SMPS measurements) and with a sampling flow range of ~ 30 – 120 L min$^{-1}$.

**Comments and suggestions:**

*Specific Comments. For Figure 2 it would be helpful to include both x ‑ and y ‑ error bars on the points, with x representing the uncertainty in the predicted pH and y the uncertainty in the pH probe/buffer measurements. This would help to know if the uncertainties include the regression line for the points that do not fall exactly on it.*

**Responses and Revisions:**

Thanks for the comments.

Both x- and y-error bars have been included in Figure 2. As shown below. Note that, since the uncertainty in the pH probe/buffer measurements were very small (with a standard deviation of ≤ 0.06), almost all the x-error bars are covered by the symbols.

[Figure]

[Figure]

[Figure]

[Figure]

**Figure 2:** Predicted pH (pH$_{predict}$) using our RGB model versus the reference pH shown on the color chart and the pH-meter-probed-pH of the buffer samples (all denoted as pH$_{reference}$) respectively, for the five different pH papers: (a) and (f) Type I: 0 – 2.5, (b) and (g) Type II: 2.5 – 4.5, (c) and (h) Type III: 4.0 – 7.0, (d) and (i) Type IV: 0.5 – 5.5 and (e) and (j) Type V: 0 – 6.0. Blue symbols denote the established relationship based on color charts only. Red symbols represent the results for 2 µL of buffer droplets on pH papers. Both vertical and horizontal error bars represent the standard deviation of five to six replicate experiments. Note that the error bars in most of the panels are smaller than the symbols.

**Comments and suggestions:**

*Specific Comments. This is a small point, but the term "outlier" is probably not the best for the point on Figure S4. If it is reproducible to the extent described it is by definition not an outlier. I think "anomalous" might be a better term, as this point would not be thrown out by the traditional Grubbs test of an outlier or other outlier tests.*

**Responses and Revisions:**

Thanks for the comments.

We have changed the term "outlier" in Figure S4 into "anomalous". And the related description in the manuscript has also been adjusted accordingly. As shown below.

[Figure]

**Figure S4.** Estimation of samples pH using the type IV pH paper. The adopted samples include a series of 2 μL lab-prepared aerosol surrogates (($NH_4$)$_2$$SO_4$-$H_2$$SO_4$, red dot) and self-prepared buffers ($Na_2$$HPO_4$-$C_6$$H_8$$O_7$, green star). pH$_{predict}$ are calculated with the averaged coefficient vector [$a$, $b$, $c$] derived from the standard buffers from three to six replicate experiments under constant photographing conditions. The error bars represent the standard deviation of three to six replicate experiments. The heights of the orange and blue bars indicate the reported pH ranges measured with pH papers and Raman spectroscopy respectively, for ($NH_4$)$_2$$SO_4$ - $H_2$$SO_4$ aerosols with particle sizes larger than 2.5 μm in Craig et al. (2018). Note that, each orange or blue bar has the same pH$_{reference}$ as of the red symbol close to it.

"*pH*$_{predict}$ versus *pH*$_{reference}$ for the 2-μL-droplet samples on the type IV pH paper are shown in Fig. S4. Generally, the *pH*$_{predict}$ by the type IV pH paper are comparable with the *pH*$_{reference}$ at a lower pH range (i.e. *pH*$_{reference}$ = 0.46, 1.52 and 3.0). However, an anomalous point (highlighted by the arrow in Fig. S4) with 1.5 unit of overestimation in *pH*$_{predict}$ can be found at *pH*$_{reference}$ around 4. ..."

**Comments and suggestions:**

*Specific Comments. The last point would be to that though the mention "anti-interference" it would be useful for the authors to see if their RGB method would work with brown carbon or black carbon samples (or some other chromophoric aerosol) that also contain secondary species and water.*

**Responses and Revisions:**

Thanks for the comments.

To further test the feasibility of our method for real aerosol samples, we generated aerosol particles under laboratory conditions and collected them on pH papers by using two custom-made impactors. In addition, the potential interference of black carbon (BC) on aerosol pH prediction was also examined. Generally, we could reasonably predict the pH of lab-generated aerosols based on the colorimetric analysis method proposed in the manuscript. The results also confirm the technical feasibility of collecting aerosol particles on pH papers through impactors. Moreover, the potential interference of BC on aerosol pH prediction was proved to be non-significant when we adopted a BC concentration representative of ambient BC levels. Details can be found in Sect. 4.2 in the revised manuscript:

**"4.2 Black carbon (BC) interference**

[revised manuscript text omitted]